

# An insight into the capability of the Actuator Line Method to resolve tip vortices

Pier Francesco Melani[1], Omar Sherif Mohamed[1], Stefano Cioni[1], Francesco Balduzzi[1], Alessandro Bianchini[1, *]

[1]Department of Industrial Engineering, University of Florence, Florence, 50139, Italy

*Correspondence to*: Alessandro Bianchini (alessandro.bianchini@unifi.it)

**Abstract.** The Actuator Line Method (ALM) is being increasingly preferred to the ubiquitous Blade Element Momentum (BEM) approach in several applications related to wind turbine simulation, thanks to the higher level of fidelity required by
the design and analysis of modern machines. Its capability to resolve the vortex-like structures shed at the blade tip (i.e., tip vortices) and their effect on the blade load profile is, however, still unsatisfying, especially when compared to other medium-fidelity methodologies such as the Lifting Line Theory (LLT). Despite the numerical strategies proposed so far to overcome this limitation, the reason for such behaviour is still unclear. To investigate this aspect, the present study uses the ALM tool developed by the authors for the ANSYS® FLUENT® solver (v. 20.2) to simulate a NACA0018 finite wing for different pitch
angles. Three different test cases were considered: high-fidelity blade-resolved CFD simulations, to be used as a benchmark, standard ALM, and ALM with the spanwise force distribution coming from blade-resolved data (*frozen ALM*). The last option was included to isolate the effect of force projection, using three different smearing functions. For the post-processing of the results, two different techniques were applied: the *LineAverage* sampling of the local angle of attack along the blade and state-of-the-art Vortex Identification Methods (VIM) to outline the blade vortex system. The analysis showed that the ALM can
account for tip effects without the need of additional corrections, provided that the correct angle of attack sampling and force projection strategies are adopted.

## 1 Introduction

### 1.1 Background and motivation

The Actuator Line Method (ALM) (Sørensen and Shen, 2002), i.e., the replacement of rotor blades with dynamically equivalent
actuator lines inside a Computational Fluid Dynamics (CFD) framework, is being increasingly preferred in wind turbine simulations to the widely used Blade Element Momentum (BEM) approach. This holds true particularly in modern, large, and highly flexible turbines (Veers et al., 2023) due to the higher level of fidelity required by their aeroelastic design and analysis (Boorsma et al., 2020; Perez-Becker et al., 2020). In fact, as the size of their rotors is progressively getting larger and larger, the study of Horizontal-Axis Wind Turbines (HAWTs) needs tools capable of resolving the interaction between the
increasingly deformable blades and the turbulent structures generated in the atmosphere at the micro- and mesoscale level, as well as in the wake of neighbouring turbines (Veers et al., 2019). In addition, a side topic in which ALM is receiving attention is connected to the renewed interest in Vertical-Axis Wind Turbines (VAWTs) for deep-sea, floating offshore installations (Cooper, 2010); the inherently unsteady aerodynamics of these machines (Ferreira et al., 2007), connected to the continuous variation of angle of attack, make the use of blade-resolved simulations prohibitive, paving the way for the use of a method
like ALM. There is still a gap, however, between this and other medium-fidelity approaches such as the Lifting Line Theory (LLT) in terms of accuracy in the resolution of the vortex-like structures shed at the blade tip (i.e., tip vortices) and their effect on the blade loads (Balduzzi et al., 2018). This issue becomes critical when simulating high-load conditions, such as high Tip-Speed Ratios (TSRs) in VAWTs, since the ALM largely overestimates the rotor power production (Melani et al., 2021b).



Upon examination of the literature, the scientific community seems to agree that the reason for this behaviour lies in the tendency of the ALM to "overspread" the computed aerodynamic forces into the domain. The resolved tip-vortex structure and the corresponding downwash along the blade are therefore underestimated. Shives and Crawford conducted the first systematic investigation on the influence of the kernel width $\beta$ and relative cell size $\beta/h$ on the predicted blade loads (Shives and Crawford, 2013). Their study revealed that, for a fixed-wing case, the parameter $\beta$ should be scaled as a fraction of the chord length, c. They demonstrated that a ratio $\beta/c$ around 0.25 is required to successfully simulate the tip vortex system within the ALM framework. To maintain an accurate estimation of the local angle of attack, which was sampled along the actuator line, they also imposed a constraint on the ALM element size, $h_{ALM} < 0.25\beta$. Three years later, Martínez-Tossas (Martínez-Tossas et al., 2016) and Jha et al. (Jha et al., 2014; Jha and Schmitz, 2018) proved that the findings of Shives and Crawford also apply to the simulation of Horizontal-axis Wind Turbines (HAWTs). Jha et al. also proposed to reduce $\beta/c$ towards the tip according to an elliptical law, suggesting this would improve the prediction of the blade circulation distribution. Taking the concept even further, Churchfield et al. (Churchfield et al., 2017) replaced the standard *isotropic* Gaussian smearing function with an arbitrary one, which can be shaped according to the actual geometry of the rotor blade. Although effective, the two methodologies require case-specific tuning and finer grids than the standard ALM, since the mesh in the rotor region needs to be refined to accommodate the resolution adopted at the blade tip. Although extensive and innovative for the time, the studies described so far are not exempt from limitations, in particular: a) the lack of proper validation data, usually replaced by analytical solutions or simulations done with low-order methods like BEM; b) the lack of insight into the physical/numerical mechanisms involved, as most analyses focused on integral quantities like the blade torque; c) the complexity of the selected test cases. Point (c) refers to the studies on horizontal-axis rotors, where the effect of blade tapering makes it more complicated to isolate the effect of trailing vorticity on the blade loads.

Recent studies from the National Renewable Energy Laboratory (NREL) (Martínez-Tossas and Meneveau, 2019) and Danmarks Tekniske Universitet (DTU) (Dağ and Sørensen, 2020; Meyer Forsting et al., 2019) took a different direction, focusing on computational efficiency rather than the accuracy of the ALM in describing the blade vortex system and its effects on the loads. The solution proposed by both institutes is a hybrid model, which corrects the over-diffusion of aerodynamic forces typical of the ALM by estimating via LLT the contribution to the *downwash* induced by tip vortexes that is dissipated in smearing of the blade forces. Although effective and robust, as also demonstrated by some of the authors in a recent publication (Melani et al., 2022), this approach has the major flaw that the induction coming from the vortices already shed in the wake must be accounted for with some sort of wake model. If this is feasible for horizontal-axis machines, at least in simple cases, it becomes almost impossible for vertical-axis ones, where the tip vortices from different blades interact with each other downstream of the turbine (Dossena et al., 2015). Therefore, the potential of the ALM method is not fully exploited.

### 1.2 Scope of the study

In this perspective, this study aims at extensively investigating the ALM's capability to simulate tip effects, using the ALM tool developed by the authors for the ANSYS® FLUENT® solver (v. 20.2) (Melani et al., 2021b). Object of the analysis was a NACA0018 finite wing, under different pitch angles. Three different test cases were considered: high-fidelity blade-resolved CFD (BR-CFD) simulations, to be used as a benchmark, ALM without any correction (*standard ALM*) and ALM with the spanwise force distribution extracted from blade-resolved data (*frozen ALM*). The last option was included to isolate the contribution of the adopted force projection strategy, using both *isotropic* and *anisotropic* Gaussian smearing functions.

For comparison of the three cases, two different families of post-processing techniques were applied. On one hand, the blade spanwise flow field was analysed via the *LineAverage* technique for the sampling of the local angle of attack (Jost et al., 2018), recently validated by some of the authors in previous work (Melani et al., 2020). In the analysis of BR-CFD results, this approach was sided by the angle of attack reconstruction technique from Soto Valle et al. (Soto-Valle et al., 2021, 2020), which is based on the analysis of the blade pressure coefficient distribution. On the other hand, the most recent Vortex Identification





Methods (VIM) from the literature (van der Wall and Richard, 2006; Soto-Valle et al., 2022) were adopted to outline the
structure and decay of the tip vortex.

**2 Test case**

For the simulation campaign, a constant-chord NACA0018 wing was selected, whose geometrical characteristics are reported
in Table 1. The choice of such a simple test case was justified by the necessity of reducing as much as possible the number of
governing parameters involved, thus increasing the generality of the analysis, and minimizing the possibility of biases. For
instance, having a constant chord c along the span ensures that the observed load reduction towards the tip was only related to
the presence of the tip vortex, without the spurious contribution of the blade tapering as in many studies on the subject ((Jha
et al., 2014; Churchfield et al., 2017; Martínez-Tossas and Meneveau, 2019; Meyer Forsting et al., 2019)).

**Table 1 Main geometrical parameters of the test wing.**

| Name | Value |
|---|---|
| airfoil | NACA0018 |
| chord c [m] | 0.382 |
| height H [m] | 3.82 |
| Aspect Ratio AR=c/H [-] | 10 |

This test wing was simulated with both blade-resolved CFD (see Section 4) and ALM (see Section 3) approaches, at the
operating conditions reported in Table 2. The freestream, chord-based Reynolds number was selected for this airfoil to obtain
a behaviour as linear as possible at the considered loading conditions, i.e., low- ($pitch=6°$), mid- ($pitch=8°$) and high-load
($pitch=10°$), without excessively raising the computational cost as it would happen in a high-Re case. The inlet Mach number
M instead was kept at a minimum to avoid undesired compressibility effects.

**Table 2 Freestream values for the simulations.**

| Name | Value |
|---|---|
| Reynolds number Re [-] | 500k |
| blade pitch [deg] | [0 2 4 6 8 10] |
| velocity $V_0$ [m/s] | 20.01717 |
| density ρ [kg/m³] | 1.18396 |
| temperature T [K] | 298.15 |
| Mach number M | 0.058 |
| turbulence intensity I [%] | 1 |
| turbulent length scale L [m] | 1 |

**3 Actuator Line Method (ALM)**

In the present study, the ALM formulation from (Melani et al., 2021b) was utilized. The code is implemented within the
ANSYS® FLUENT® solver (v. 20.2), using a User Defined Function (UDF). In the ALM, the blade geometry is not directly
resolved but modeled using a lumped-parameter approach, resulting in a significant computational cost reduction. The flow
field across the rotor is resolved using CFD. The corresponding algorithm could be described as follows: firstly, the flow field



local to the actuator line is sampled, then, based on the sampled flow field, the lift and drag coefficients are obtained from tabulated polar data. Finally, the forces are projected into the CFD domain via a Gaussian function also known as *Regularization kernel*. For the sampling of the angle of attack, the code uses the novel *LineAverage* technique (Melani et al., 2021b). Originally derived from Jost et al. (Jost et al., 2018), this method calculates the undisturbed velocity $V$ as the integral average of the flow velocity field along a circular sampling line (see Eq. 9). The line is centered at the airfoil quarter-chord

and has a radius $r_s=1c$.

In this study, a variation of the ALM called the *Frozen ALM* has been utilized alongside the standard ALM. Unlike the conventional ALM, which relies on tabulated airfoil polar data to calculate aerodynamic coefficients, the *Frozen ALM* directly obtains these coefficients from a blade-resolved CFD simulation. Bypassing the use of tabulated data, the *Frozen ALM* effectively eliminates uncertainties associated with the quality of airfoil polar data and the ad-hoc correction models that are

typically employed. The *Frozen ALM* was initially introduced by (Martínez-Tossas et al., 2017) and has been previously employed by the authors in their recent work (Mohamed et al., 2022) to investigate the limitations and challenges of the ALM specifically for vertical axis turbines. Building upon this prior research, the present study applies the *Frozen ALM* for the first time in a three-dimensional flow regime. By extending the application of the Frozen ALM to this flow regime, the study aims to get a better understanding of the capabilities and potential of the ALM in accurately representing the complex flow

phenomena connected to tip vortices.

### 3.1 Regularization kernel

For the Regularization kernel, three distinct Gaussian function, shown in Fig. 1, were utilized:

1. *Isotropic Gaussian* (Shives and Crawford, 2013), at which the forces are equally distributed in the chord and thickness direction within a cylindrical shape. The expression for this function is as follows:

$$\eta(r) = \frac{1}{\beta^2 \pi} exp\left[-\left(\frac{|r|}{\beta}\right)^2\right] \qquad (1)$$

2. *Anisotropic Gaussian* (Churchfield et al., 2017), a two-dimensional distribution, wherein forces are distributed in an elliptical shape using distinct kernel widths in the chord and thickness directions, denoted as $\beta_c$ and $\beta_t$, respectively. The expression for this function is:

$$\eta(r_c, r_t) = \frac{1}{\beta_c \sqrt{\pi}} exp\left[-\left(\frac{|r_c|}{\beta_c}\right)^2\right] \cdot \frac{1}{\beta_t \sqrt{\pi}} exp\left[-\left(\frac{|r_t|}{\beta_t}\right)^2\right] \qquad (2)$$

3. *Anisotropic Gaussian-Gumbel* (Schollenberger et al., 2020), akin to the anisotropic Gaussian, but with an incorporated Gumbel function in the chordwise direction, mimicking the airfoil shape. The expression for this function is:

$$\eta(r_c, r_t) = \frac{1}{\beta_c} exp\left[-\left(\frac{|r_c|}{\beta_c}\right)\right] \cdot exp\left[-exp\left[-\left(\frac{|r_c|}{\beta_c}\right)\right]\right] \cdot \frac{1}{\beta_t \sqrt{\pi}} exp\left[-\left(\frac{|r_t|}{\beta_t}\right)^2\right] \qquad (3)$$


In Eqs. (1-3), $r$ denotes the distance between the centroid of the generic cell and the actuator line, while $\beta$ represents the kernel width parameter, which is associated with a characteristic dimension of the airfoil (e.g., chord, $c$), as described in (Shives and Crawford, 2013). For the *isotropic Gaussian*, $\beta$ was set to 0.1c, while for both the *anisotropic Gaussian* and *Anisotropic Gaussian Gumbel*, it was defined as 0.2c in the chord-wise direction, $\beta_c$, and 0.1c in the thickness-wise direction, $\beta_t$, These

specific values were selected based on the calibration analysis conducted by the authors in Mohamed et al. (Mohamed et al., 2022) for a two-dimensional airfoil. For more comprehensive information about the ALM code, please refer to (Melani et al., 2021b).



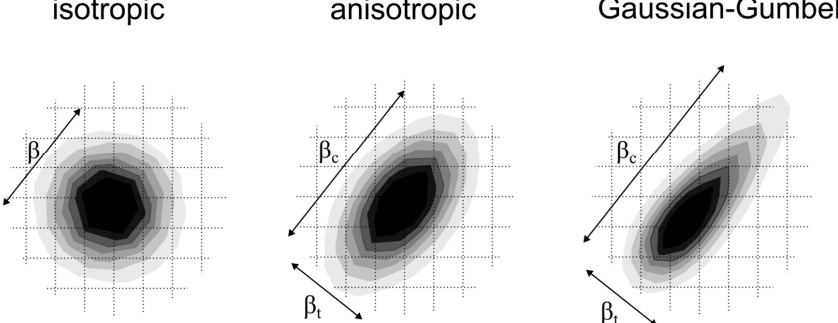

**Figure 1 Schematic representation of the three different kernel functions used for ALM simulations in the present work.**

**3.2 Numerical set-up**

ALM simulations were carried out with the steady Reynolds-Averaged Navier-Stokes (RANS) CFD solver available in ANSYS® Fluent® (v. 20.2). The corresponding setup followed a consolidated numerical approach developed by some of the authors for airfoil simulation (Balduzzi et al., 2021), which features the *coupled* algorithm for pressure-velocity coupling, and the 2nd order upwind scheme for both RANS and turbulence equations. The *k-ω SST* is used for turbulence modelling.

Figures 2a and 3a illustrate the adopted computational domain, whose dimensions – length L = 60c, width W = 40c and height $H_D$= 3H – were selected to minimize blockage effects and allow the blade wake to properly develop. At the boundaries, the standard *far-field* boundary conditions for external flows are applied: uniform velocity at the inlet, ambient pressure at the outlet, and *symmetry* on the other surfaces, including the bottom one. This way, the spanwise symmetry of the problem was exploited, thus halving the global number of elements.


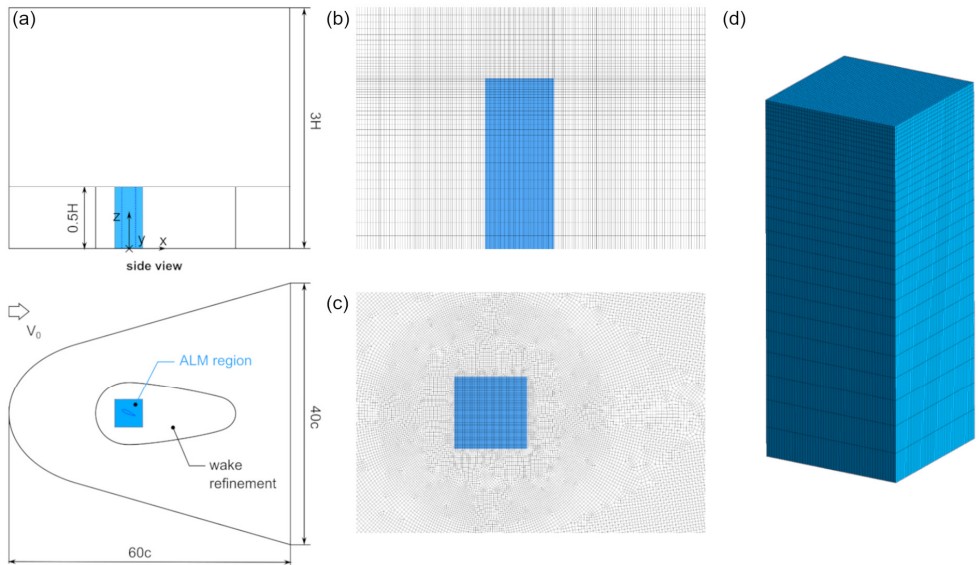

**Figure 2 Overview of the grid used for ALM simulations: a) computational domain b) side view of the mesh in correspondence of the wing c) detail of the ALM region at the wing midspan d) detail of the surface mesh along the ALM region in the spanwise direction.**




The discretization strategy was selected to ensure the optimal resolution of the vorticity distribution along the blade span as well as in the wake, i.e., the tip vortex, at the minimum computational cost, following the results of a sensitivity analysis performed by some of the authors in a previous work (Melani et al., 2022). Details of this setup are reported in Table 3.

A uniform cartesian grid was used for the ALM region (see Fig. 2c), as required by the ALM method, adjusting the corresponding grid resolution to the local kernel size $\beta/c$ under the constraint $h_{ALM} < 0.4 \cdot \min(\beta_t, \beta_c)$ for stability reasons. The

criterion adopted for the dimensioning of the smearing radius $\beta$ varied with the kernel shape. For the standard *isotropic* function, a value of $\beta c = \beta t = 0.1c$, ensuring a correct description of the tip vortex core, was selected (Melani et al., 2022). The *anisotropic* and *Gauss-Gumbel* functions were tuned instead to minimize the error in terms of velocity field with respect to BR-CFD in a two-dimensional environment (Mohamed et al., 2022). As this process took place without considering tip effects, the setup of these two functions might not be optimal for the scope of this study. Therefore, possible conclusions about their

application must be verified in future work. The ALM region was progressively expanded to the domain boundaries via an unstructured, quadrilateral mesh, always scaling the local cell size on $h_{ALM}$. The bottom grid was then extruded along the blade span, optimizing the grid density at the tip by distributing the elements according to an exponential bias, as shown in Fig. 2b-e (Melani et al., 2022). For further details, please refer to (Melani et al., 2021b).

**Table 3 Characteristics of the different grids used for BR-CFD and ALM simulations in the present work.**

| Name | blade-resolved CFD | ALM - iso | ALM - aniso | ALM - GG |
|---|---|---|---|---|
| # elements sliding interface | 500 | | 500 | |
| #elements (blade) | 720 | | - | |
| # layers (boundary layer) | 40 | | - | |
| # elements span | 150 | | 30 | |
| spanwise bias factor [-] | - | | 1.08 | |
| thick-wise kernel size $\beta_t/c$ [-] | - | | 0.1 | |
| chordwise kernel size $\beta_c/c$ [-] | - | 0.1 | 0.2 | 0.2 |
| # cells [$10^6$] | 44.1 | | 4.35 | |

## 4 Blade-resolved CFD

Blade-resolved CFD simulations employed the same numerical schemes, computational domain, and meshing strategy of the ALM ones (see Section 3.2). Important differences, visible in Figs. 2 and 3 and reported in Table 3, exist though in the wing region due to the presence of the airfoil geometry. According to the experience of the authors on similar test cases (Balduzzi

et al., 2021), the blade surface was modelled as a smooth *no-slip* wall, discretized with an O–type grid of 720 quadrilateral elements. To ensure a proper resolution of the boundary layer, a dimensionless wall distance ($y^+$) lower than ~ 1 and a total number of layers of 40 were employed in the direction normal to the wall (see Fig. 3e). Given the Reynolds number used in the tests (see Table 2), an intermittency transport equation was added to the *k-ω SST* model to include the effects of turbulent transition.

The wing region was connected to the far-field domain, this time discretized with an unstructured, triangular mesh, via a *sliding interface* (see Fig. 3d), so it could be rotated of the imposed blade pitch angle. The final, three-dimensional mesh was once again obtained by extruding the bottom one along the blade span. In this case, the cells were distributed not according to an exponential bias like in the ALM but assigned to the cell blocks of variable height from midspan to tip (see Fig. 3b-e). This strategy ensured better control of the aspect ratio of the elements along the wing surface, preventing in particular the excessive

stretching of those at the midspan (Balduzzi et al., 2017).





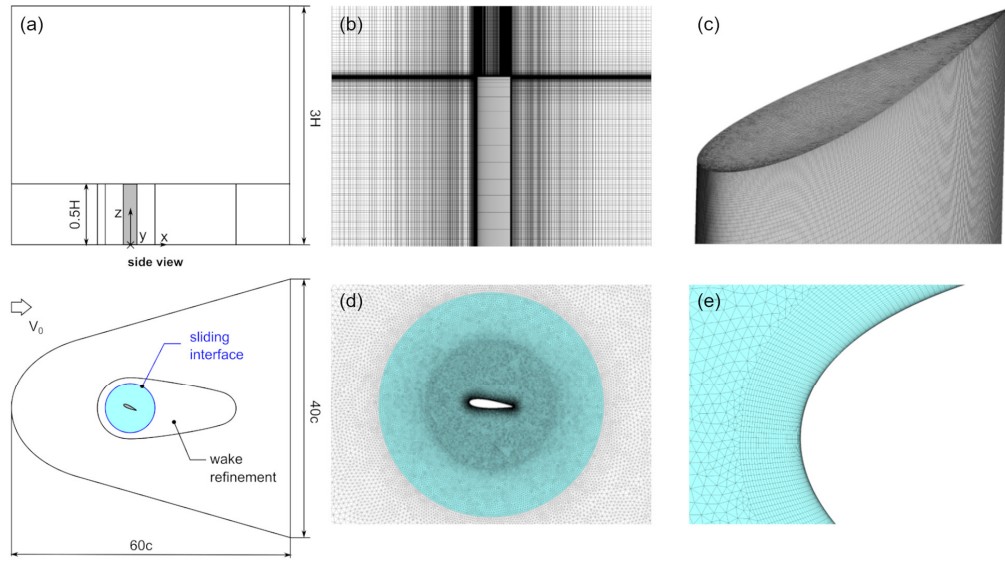

**Figure 3 Overview of the grid used for blade-resolved simulations: a) computational domain b) side view of the mesh in correspondence of the wing c) detail of the surface mesh at the wing tip d) detail of the rotating region at the wing midspan e) detail of the prismatic grid used for the boundary layer discretization at the blade leading edge.**

This setup has been validated, at least for the two-dimensional case, against the experimental measurements of Timmer (Timmer, 2008), as shown in Fig. 4. It is observed how the matching between 2D numerical results and experiments is good until AoA=10°, with a nearly perfect matching in terms of slope of the lift curve and drag value at the zero-lift point. Approaching the static stall point, the two datasets start diverging, as the stall predicted by BR-CFD occurs earlier than the measured one. This issue does not affect the present study, as the analysis is limited to the attached flow region, with a maximum tested angle of attack of 10° (see Table 2), thus justifying the use of 2D BR-CFD polar data for both the reconstruction of the equivalent angle of attack from BR-CFD simulations of the finite wing (see Section 6.1) and for standard ALM simulations (see Section 6.3).

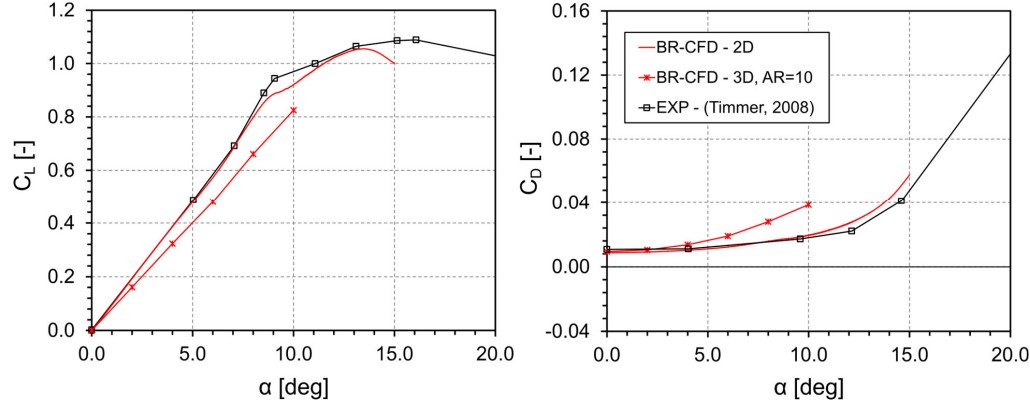

**Figure 4 Comparison in terms of lift ($C_L$) and drag ($C_D$) coefficients between BR-CFD simulations, both 2D and 3D, and the experimental measurements from (Timmer, 2008), for Re=500e3.**



On the other hand, the accuracy of 3D BR-CFD simulations could not be verified, as no reliable data for this test case is currently available in the literature. The confidence in the adoption of these results derives indeed from the experience of some of the authors in this kind of simulations (Balduzzi et al., 2017) and from the nature of the current analysis, whose aim is not to provide a benchmark for this test case but rather some insights into the physical mechanism underlying tip losses and the capability of the ALM to reproduce it.

## 5 Data postprocessing

Two different families of post-processing techniques were applied to the results of both BR-CFD and ALM simulations. On one hand, the most recent Vortex Identification Methods (VIM), described in section 5.1, were adopted to outline the structure and decay of the tip vortex. On the other hand, the blade spanwise flow field was analysed via the *LineAverage* technique for the sampling of the local angle of attack, whose details can be found in Section 5.2.

### 5.1 Tip vortex tracking metrics

The various modelling strategies analysed in this work affect in turn the tip vortices shed from the airfoil. Hence, the effect of the applied methodology on the vortex structure was investigated through multiple metrics, namely vortex centre position, core radius and circulation. These properties are affected by viscous decay, as the vortex is convected downstream, so the analysis is performed over four vertical sampling planes with varying distance from the airfoil (1, 2 and 5 chords), starting from the aerodynamic centre (see Fig. 5).

As summarized in Fig. 5, the tip vortex metrics were computed following the methodology commonly used in the literature (van der Wall and Richard, 2006; Soto-Valle et al., 2022):

- *vortex center C*: the vortex center defines the axis of rotation of the vortical structure, and it is used to track the position of the tip vortex as it is convected downstream. In the present work, its position is computed from the resolved velocity field by calculating the $\lambda_2$ scalar field (Jeong and Hussain, 1995), and locating the position of its minimum on the sampling plane. This methodology was selected among the available vortex identification methods as it can distinguish the contributions of viscous stresses and irrotational straining;

- *vortex core radius $r_C$*: the core radius defines the size of the inner part of the vortex, where the fluid rotates as a rigid body. As the tip vortex is convected downstream, the vortex structure is dissipated due to viscous decay and the size of the core increases (*vortex aging*). In this work, the core radius is calculated from the induced velocity field as the distance between the vortex center C and the location of maximum induced velocity $V_{ind}$ (Mauz et al., 2019) (see Fig. 5). $V_{ind}$ is calculated by subtracting the convection velocity $(u_c, v_c)$ of the vortex from the velocity field,

$$u_{ind} = u - u_c \tag{4}$$
$$v_{ind} = v - v_c \tag{5}$$

The convection velocity is assumed equal to the velocity in the vortex center, where the vortex induced velocity is zero (Yamauchi et al., 1999; van der Wall and Richard, 2006):

$$u_c = u(x_c, y_c) \tag{6}$$
$$v_c = v(x_c, y_c) \tag{7}$$

The core radius is calculated from horizontal and vertical slices of the induced velocity field, passing by the vortex center. In this way, the velocity profile of the vortex (see Fig. 5) is obtained along the two directions. The results are averaged to provide a more representative value. Additionally, the Aspect Ratio AR of the vortex is calculated to account for possible asymmetry of the vortical structure;

- *vortex circulation $\Gamma$*: circulation is a measure of the vortex intensity and is used alongside the core radius to measure its aging in the wake. In this work, it was computed as the integral of the in-plane vorticity $\omega_x$. In order to avoid the inclusion



of spurious contributions, the integration domain was centered on the vortex center and limited to a radius of 2c from the vortex center.


$$\Gamma = \int_A \omega dA \qquad (8)$$

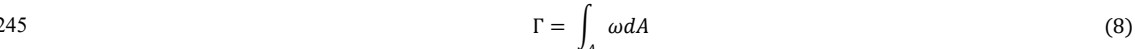

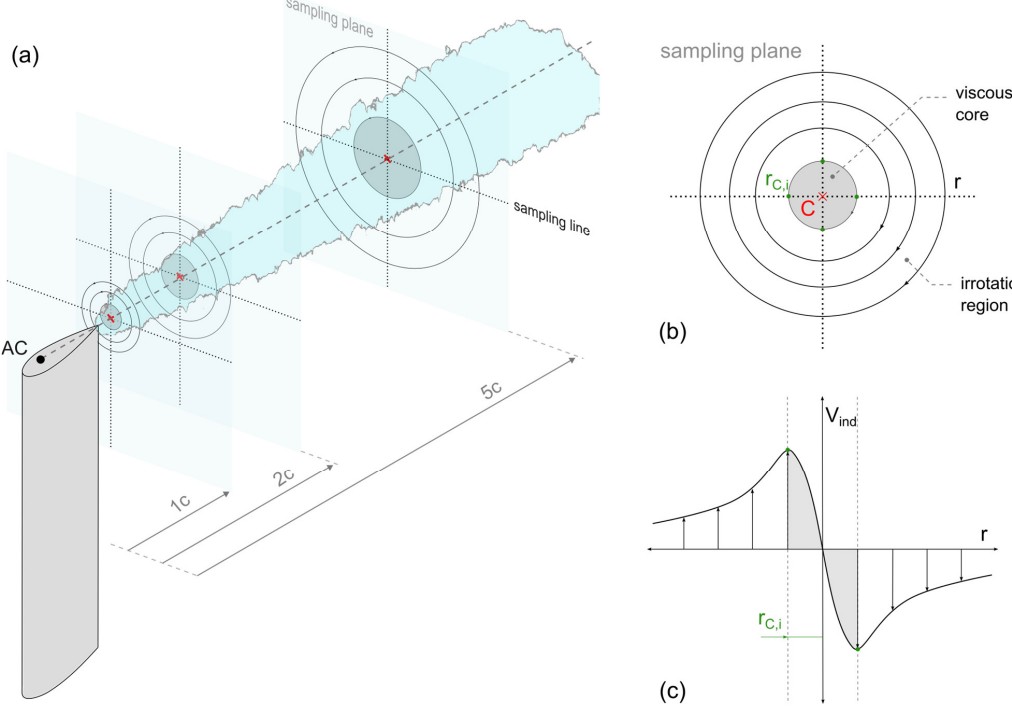

**Figure 5 Schematic representation of the sampling setup used in the present work: (a) sampling planes b) vortex sketch and definition of core radius c) velocity profile of the vortex.**

**5.2 Angle of attack sampling**

The *LineAverage* method has been originally introduced by Jost et al. (Jost et al., 2018) for HAWTs, in the attempt of increasing the accuracy of previously available methods in capturing the effects of shed and trailing vorticity on the measured angle of attack on turbine blades. To this end, the undisturbed velocity V is computed as the integral average of the flow velocity field along a closed line around the airfoil (see Fig. 6):

$$V = \frac{\sum_{j=1}^{N} \vec{v_j} \cdot |\vec{s_j}|}{\sum_{j=1}^{N} |\vec{s_j}|} \qquad (9)$$

where $\vec{v_j}$ is the local velocity and $\vec{s_j}$ the arc length at the node j along the sampling circle. According to its creators, this sampling strategy should be able to completely remove the effect of bound circulation on the local inflow velocity, since in the averaging process the induced velocity components on any pair of opposite points on the closed path are levelled out, still accounting for the net distortion associated with trailing vorticity.

The choice of this method for the present investigation was justified by the accuracy and robustness shown in: (i) a previous

work of some of the authors on blade-resolved simulations of vertical-axis wind turbines (Melani et al., 2020), in which it has been validated against high-fidelity numerical and experimental data; (ii) its application to ALM simulations of both horizontal- (Bergua et al., 2023) and vertical-axis machines (Melani et al., 2021b, a).



The analysis carried out in the present work used a sampling radius $r_s$ = 1c and N=80 evenly distributed sampling points, as recommended by (Jost et al., 2018; Rahimi et al., 2018).




**Figure 6 Schematic representation of the sampling setup used in the present work.**

## 6 Results

In this Section, the main outcomes of the investigation are presented. In Section 6.1, a deeper insight into the physical
mechanism responsible for the load degradation observed in the reference blade-resolved simulations is provided. Although partially redundant with respect to what is found in the literature, this passage is considered fundamental by the authors to give rigor and consistency to the following analyses on the Actuator Line Method. These are carried out in Section 6.2. The first part focuses on the *frozen ALM*, i.e., the insertion in the computational domain of the blade forces coming from BR-CFD. The second one completes the analysis by comparing the results of the ALM, in its standard formulation, with those obtained with
the *frozen ALM* strategy and the BR-CFD reference data. This way, the error introduced by adopting a version of the ALM, which is not tailored to the resolution of tip vortexes, could be quantified.

### 6.1 Investigation of the flow mechanism

A preliminary and fundamental step of the present investigation was the identification of the physical mechanism underlying the spanwise load degradation observed in the blade-resolved (BR-CFD) simulations taken as reference. It was a priority
indeed to understand if this mechanism actually falls in the spectrum of the aerodynamic phenomena interpretable as a dynamically equivalent variation of a two-dimensional angle of attack, as prescribed by the Lifting Line Theory (LLT) (Prandtl and Tietjens, 1934), or if it is characterized by inherent three-dimensional characteristics. Only in the first case, in fact, the ALM would have the possibility to capture the effect of tip vortices on the blade loads without the need for semi-empirical corrections, as it is based on the use of tabulated polar data. This work would then have a solid theoretical foundation. In the
second case instead, the only way to improve the ALM accuracy would be to rely on dedicated corrections such as the ones currently used for the simulation of wind turbines, e.g., (Glauert, 1935; Shen et al., 2014).

For this purpose, a two-dimensional angle of attack $\alpha_{2D, eq}$ was computed for each spanwise section of the blade. As for blade-resolved simulations, it is not possible to use the *LineAverage* method (see Section 5.2) for radii lower than the airfoil chord, since the sampling line would intersect the airfoil surface, a different strategy was adopted, following the work of Soto Valle
et al. (Soto-Valle et al., 2021, 2020). More in detail, $\alpha_{2D, eq}$ was selected as the angle of attack that, when imposed to a two-dimensional calculation, would minimize the error of the predicted pressure coefficient $C_P$ (Eq. (10)) distribution along the chord with respect to the one of 3D BR-CFD. It was priority in this process to match the minimum $C_P$ value on the Suction Side (SS), as in the attached flow regime it is the main variable regulating lift production.



$$C_P = \frac{P - P_0}{\frac{1}{2}\rho V_0^2}$$

(10)

In this case, the viscid panel method used in the original approach of Soto-Valle et al. was replaced with 2D BR-CFD, using
the same airfoil meshing strategy and turbulence modelling of 3D simulations (see Section 4). In this way, the bias in the computation of $\alpha_{2D, eq}$, related to the inherent differences between CFD and panel methods, is avoided.

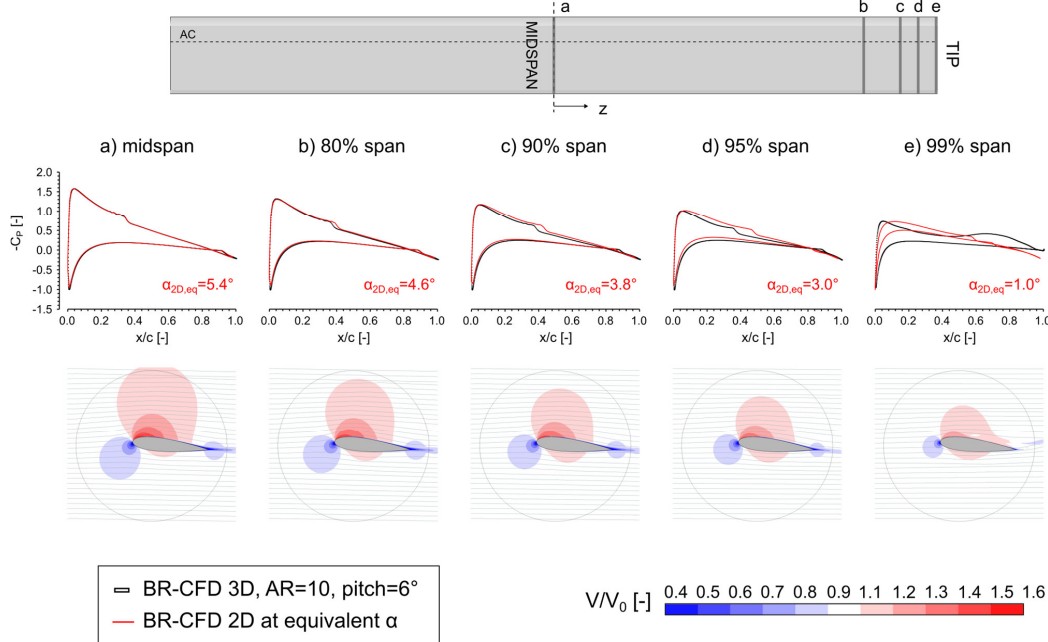

**Figure 7** Extraction of the pressure coefficient $C_P$ profile and non-dimensional velocity field $V/V_0$ from 3D BR-CFD from different
sections along the blade span, for pitch=6°. 3D pressure data is compared with those coming from 2D BR-CFD simulations for the computation of the equivalent angle of attack $\alpha_{2D, eq}$.

Figure 7 presents the results of the workflow described above, along with the corresponding velocity field, for the low-load case (*pitch=6°*). For the sake of brevity, only a few relevant blade sections are shown in the picture. The spanwise load degradation associated with the tip vortex can be reasonably approximated with a reduction of the equivalent 2D angle of
attack $\alpha_{2D, eq}$ until 80% of the blade span. Consequently, the deviation between the section lift coefficient computed $\alpha_{2D, eq}$ and the 3D one is limited, as shown in Fig. 9. Therefore, this flow region takes here the name of "*2D region*". This effect is also visible from the non-dimensional velocity field, with the occurrence of a progressive shift of the stagnation point towards the center of the airfoil leading edge and a reduction of the SS suction peak.

The same trend is found between 80 % and 97 % of the span. Going towards the tip, however, a pressure deviation of increasing
intensity arises between 3D and 2D BR-CFD in the rear of the blade ($0.5 \leq x/c \leq 0.9$), despite the good correspondence between the two datasets in the leading edge region. This phenomenon, called "*decambering effect*", is well-known in the literature (Sørensen et al., 2016) and derives from the radial outflow generated by the tip vortex structure. Traces of this pattern are visible in Figs. 7d-e and 8d-e as a deviation of the freestream velocity vectors towards the SS wall. As in this part of the blade the flow progressively becomes three-dimensional and the deviation between 2D and 3D lift coefficients is increasing
(see Fig. 9), the corresponding flow regime takes the name of "*transition region*".

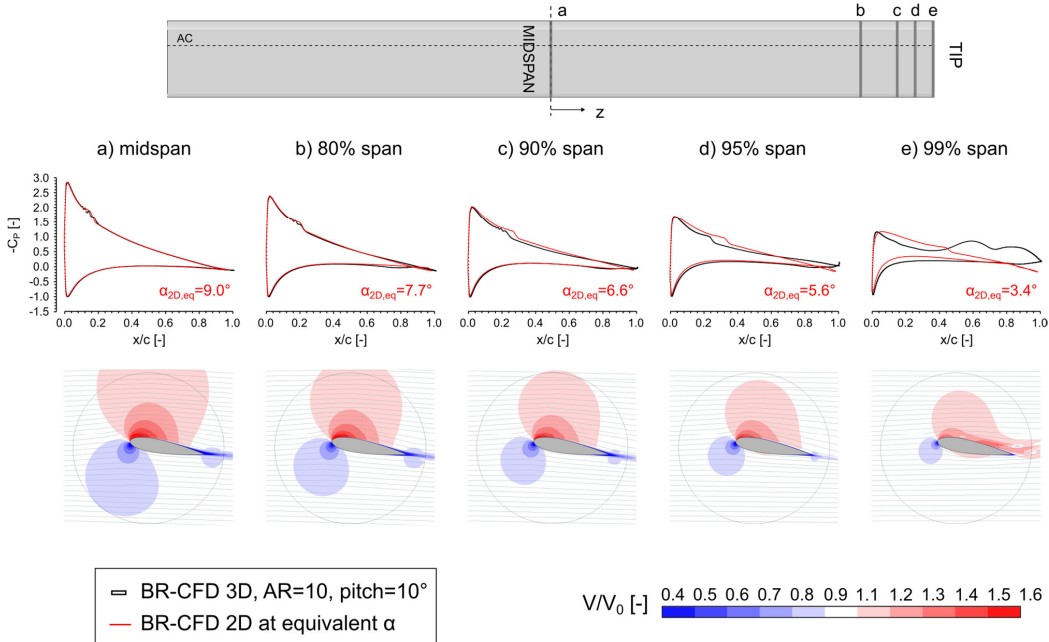

**Figure 8** Extraction of the pressure coefficient $C_P$ profile and non-dimensional velocity field $V/V_0$ from 3D BR-CFD from different sections along the blade span, for pitch=10°. 3D pressure data is compared with those coming from 2D BR-CFD simulations for the computation of the equivalent angle of attack $\alpha_{2D, eq}$.

In the region between 97% of the span and the tip instead, the flow becomes fully three-dimensional (*"3D region"*), so that 2D theory and the concept of angle of attack itself lose validity (Branlard, 2017). Indeed, it is not possible anymore to approximate the blade loads using polar data, as testified in Fig. 9. The extension of this region and the intensity of the corresponding loads are small enough though for the hypothesis of an $\alpha_{2D, eq}$ to be still adopted, confirming the prescriptions of the Lifting Line Theory.


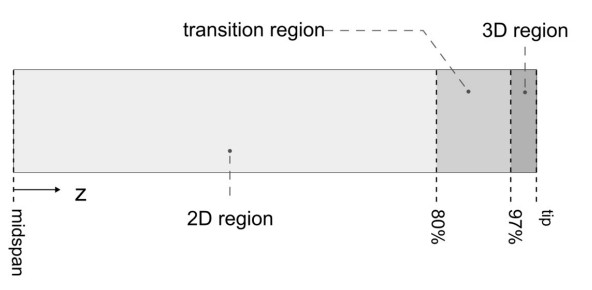
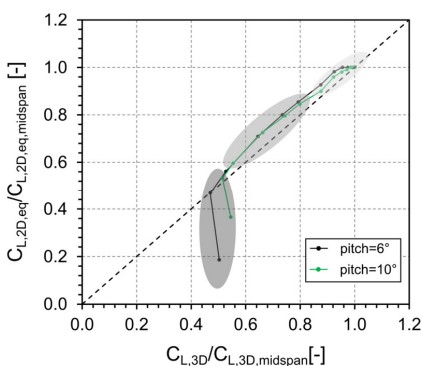

**Figure 9** Normalized plot of the lift coefficient computed from $\alpha_{2D, eq}$ vs the one directly extracted from 3D BR-CFD simulations. The amount of deviation from the ideal situation in which the two are the same is used to identify the different flow regimes developing along the blade span.

As the same considerations can be made for the high-load scenario (*pitch=10°*) presented in Fig. 8 without losing coherence, it is possible to conclude, at least for this specific case, that the effect of tip vortices on a resolved flow field can be reasonably approximated as an angle of attack reduction. A corollary of this conclusion is that the ALM can, in theory, reproduce this effect without resorting to additional corrections. Whether this is actually feasible or not will be investigated in Section 6.2. A



feature of the blade spanwise flow that is apparent from Figs. 7 and 8 and should not be ignored is that the reduction of $\alpha_{2D,\,eq}$

does not happen in the freestream, which remains basically undisturbed, but at the *blade chord scale*. This is key when deciding the velocity sampling strategy, as later explained in Section 6.2.2.

## 6.2 ALM

As discussed, Section 6.1 established that, at least for this test wing, the main mechanism responsible for the spanwise load degradation observed in the BR-CFD results is a progressive reduction of the local equivalent angle of attack $\alpha_{2D,\,eq}$ along the

blade. This confirmed the assumptions of Lifting Line Theory. Starting from there, the question naturally arose whether the ALM can reproduce the flow field observed in BR-CFD simulations, both along the blade span and in the wake, and, if the answer is positive, how this information can be properly extracted from the resolved velocity field for the computation of blade loads.

### 6.2.1 Blade loads

The analysis starts from the benchmarking of the *LineAverage* method (see Section 5.2) in its standard setup ($r_s$=1c), i.e., the one that is commonly found in the literature and provides good results for two-dimensional flows. The angle of attack sampled from the flow field with such method will from now on be simply referred to as α and must be distinguished from the equivalent two-dimensional angle of attack $\alpha_{2D,\,eq}$ that was extracted from BR-CFD simulations by comparing the airfoil pressure data.

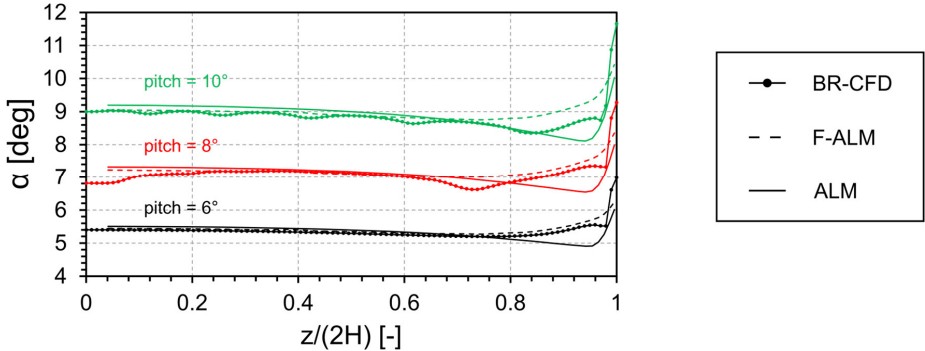


**Figure 10 Comparison between BR-CFD, frozen ALM (F-ALM) and standard ALM for the three operating conditions under consideration in terms of angle of attack sampled with the standard *LineAverage* setup (r$_s$=1c). As the three kernel shapes yield the same results, only the data for the isotropic function is reported.**

Figure 10 reports the α spanwise profile for BR-CFD, Frozen ALM (from here on abbreviated as F-ALM) and standard ALM,

for the three load conditions considered in this work, i.e., low- (*pitch=6°*), mid- (*pitch=8°*) and high-load (*pitch=10°*). For the sake of clarity, only the curves of the *isotropic* case are reported, as changing the kernel shape provided no relevant difference. The three datasets are in good agreement with each other, although both standard ALM and F-ALM tend to filter out some of the flow oscillations observed in BR-CFD simulations, and they predict an angle of attack that increases going towards the tip. As this trend does not follow the one observed in Section 6.1 and, more in general, the common understanding of the

phenomenon, relevant discrepancies arise when the sampled angle of attack is cross-compared with the corresponding blade forces (see Fig. 11). The cross-wise force coefficient $C_y$ predicted by the standard ALM, which scales linearly with α, starts deviating from the 3D BR-CFD one - computed by integrating the blade pressure distribution - already at 60 % of the span, keeping a constant profile before increasing at 90% of the span. BR-CFD simulations show instead a decreasing trend up to the tip of the blade. Looking at the streamwise force coefficient $C_x$, the difference seems smaller for most of the blade span,

but only because the drag coefficient $C_D$ is approximately constant in the attached flow region (see Fig. 4).





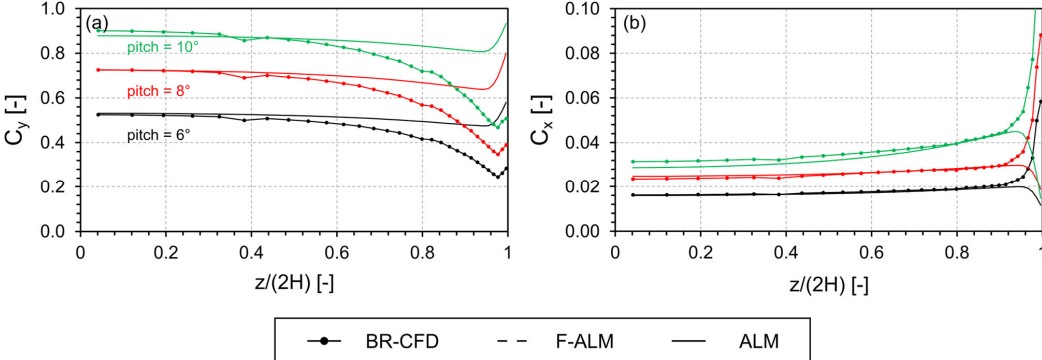

**Figure 11 Comparison between BR-CFD, frozen ALM (F-ALM) and standard ALM for the three operating conditions under consideration in terms of: a) force coefficient along the cross-wise direction; b) force coefficient along the streamwise direction. As the three kernel shapes yield the same results, only the data for the isotropic function is reported.**

At the tip, where the flow is fully three-dimensional (see Section 6.1), the 3D BR-CFD and ALM profiles deviate abruptly due to the *induced drag* associated with the shedding of the tip vortex. This phenomenon lies though outside of the range of validity of 2D airfoil theory, upon which the ALM and all methods based on polar data are built, and therefore outside of the scope of the present study.

### 6.2.2 Frozen ALM - Spanwise flow

To gain a deeper insight into the apparently unphysical behavior observed in Section 6.2.1 and, more in general, into how the flow field along the blade develops for the different simulation methods, Figs. 12 and 13 report the comparison in terms of non-dimensional velocity $V/V_0$ between BR-CFD and frozen ALM (F-ALM) along different spanwise locations, for *pitch=6°* and *pitch=10°*, respectively. The standard *LineAverage* sampling line at $r_s=1c$ is also reported for clarity. The use of the *frozen ALM* strategy (see Section 3) is justified here by its ability to remove the uncertainty related to loads computation and isolate the effect of ALM on the resolved flow field. To make the analysis as general as possible, three different kernel shapes from the literature are considered: the standard *isotropic* Gaussian and two new formulations, namely *anisotropic* Gaussian and *Gauss-Gumbel* (see Section 3.2 for details on the implementation).

It is evident how the F-ALM, given the same force distribution of blade-resolved simulations, produces a distortion of the velocity field along the span similar, at least qualitatively and with all the well-known limitations of the ALM method, to that of BR-CFD, regardless of the selected kernel shape. Even the SS high-velocity bubble's progressive elongation, related to the presence of the tip vortex, is captured. The only relevant deviation is visible at 99% of the span, where the flow becomes fully three-dimensional, thus invalidating the assumption of 2D sectional flow that underlies all polar-based methods like the ALM (see Section 6.1). However, the magnitude of this velocity distortion is underestimated with respect to BR-CFD, especially in the sections close to the blade tip, e.g., 90% and 95% of the span and with non-conventional kernel shapes such as *anisotropic* and *Gauss-Gumbel*, since they reduce the intensity of the local circulation by spreading the forces over a wider area compared to the *isotropic* formulation. This issue is well-known in the scientific community (Jha et al., 2014; Dağ and Sørensen, 2020; Martínez-Tossas and Meneveau, 2019) and has been justified by the loss of circulation related to the force spreading into the computational grid. The presented results further confirmed this theory and will be key in Section 6.2.4 to discuss the relationship between force spreading and angle of attack sampling when it comes to resolving tip vortices with the ALM.

As BR-CFD and F-ALM produce the same kind of flow distortion along the span, the angle of attack issue can be tackled by looking at the flow field alone. From Figs. 12 and 13, the reason for the behavior observed in Fig. 10 already emerges. In fact, at $r_s=1c$ the *LineAverage* sampling line lies in a relatively undisturbed flow region. If for the *midspan* section, this strategy works as intended (*LineAverage* was indeed first applied in ALM for wind turbine simulation, to capture the local deceleration





of the flow without including the contribution of bound vorticity (Melani et al., 2021a)), for the tip region, it misses most of the downwash induced by the shed vortex, yielding unphysical values for the angle of attack. As demonstrated in Section 6.1, the induction process happens indeed at the *blade chord* scale.

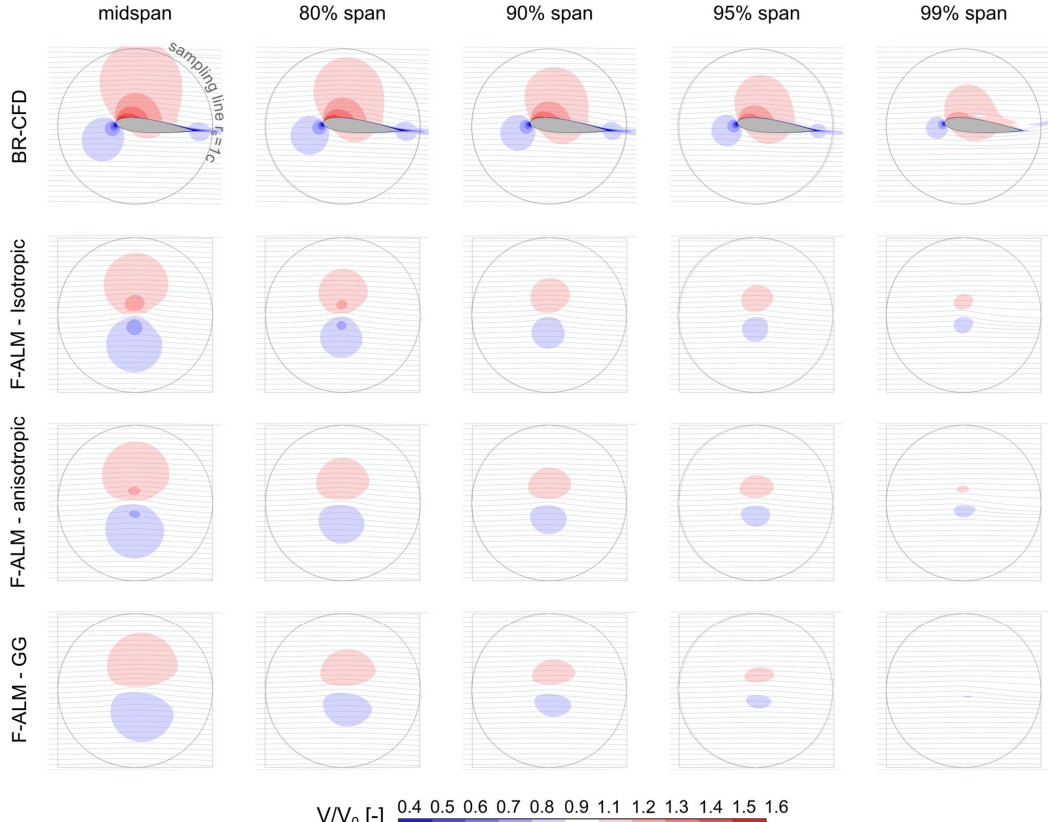

**Figure 12 Comparison in terms of non-dimensional velocity magnitude V/V₀ between BR-CFD and Frozen ALM (F-ALM) at different spanwise sections, for pitch=6°.**

To fully highlight the issues related to the velocity sampling in the near-field and start exploring possible solutions, Fig. 14 reports the comparison between BR-CFD and F-ALM in terms of non-dimensional cross-wise velocity $V_y/V_0$, which, as the freestream is purely axial, can be directly interpreted as *downwash* velocity and thus be used to track the tip vortex effect on

the blade plane. The analysis is limited here to the high-load case (*pitch=10°*) for the sake of brevity. Focusing at first on the midspan plane, it is observed how the induced velocity fields predicted by BR-CFD and F-ALM are similar in the near-field and resemble the symmetric one produced by a Lamb-Oseen vortex. This is in line with the classic ALM theory (Shives and Crawford, 2013). Moving towards the tip, the symmetry in the downwash found at the midspan is progressively broken, as the airfoil bound circulation, which is connected to the generated lift, decreases, while the tip vortex provokes the stretching of

the downwash region in the rear part of the airfoil towards the wake. This is visible from both BR-CFD and F-ALM simulations, although the F-ALM underestimates the extension of the rear induction region and the corresponding severity of the downwash gradient, as a consequence of the tendency of the ALM to "overspread" the aerodynamic forces commented in the previous paragraph. Switching to non-standard kernel shapes, something counter-intuitive happens. In fact, the strength of the tip vortex is similar for all kernel shapes, since it depends on the magnitude of the inserted forces (this aspect can be

quantitatively seen in Fig. 17), while the intensity of the local circulation reduces going from the *isotropic* to the *Gauss-Gumbel*





formulation. Consequently, the asymmetry in the downwash distribution at the Aerodynamic Center is accentuated, especially with the *Gauss-Gumbel* function.

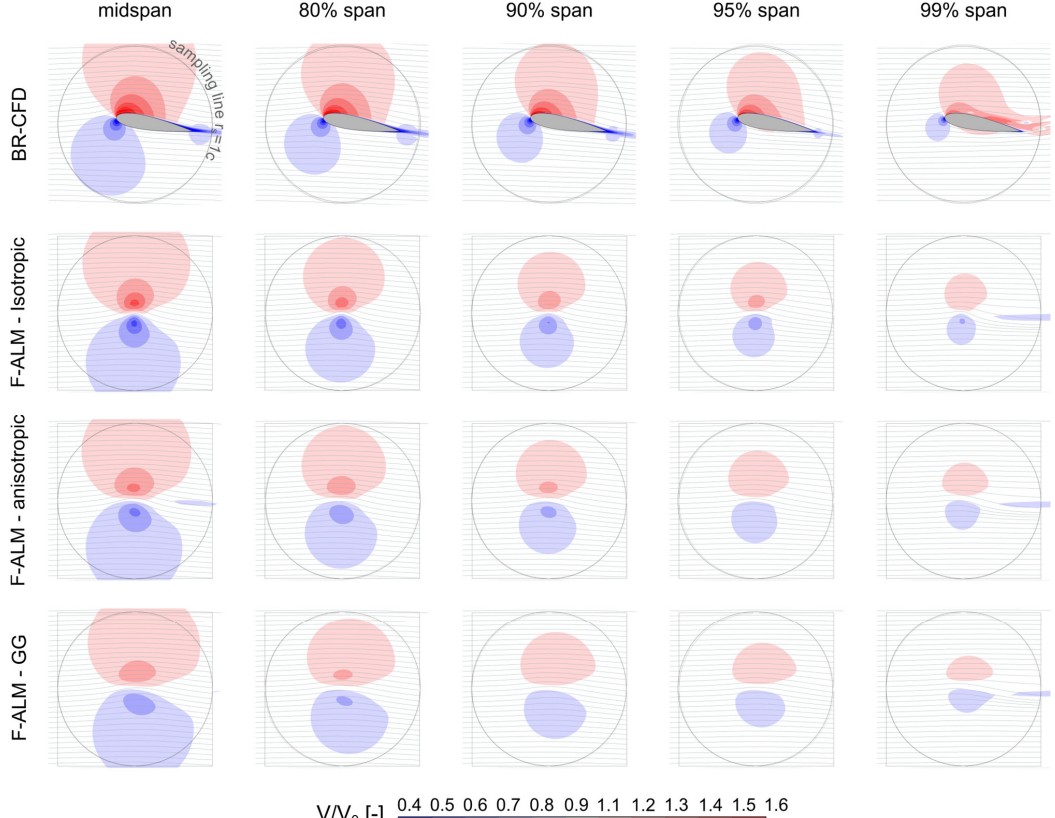

**Figure 13 Comparison in terms of non-dimensional velocity magnitude V/V₀ between BR-CFD and Frozen ALM (F-ALM) at different spanwise sections, for pitch=10°.**

The natural conclusion of the considerations made so far is that, in both BR-CFD and F-ALM, the tip vortex manifests itself as an asymmetry in the downwash velocity on the blade plane, increasing in intensity going towards the tip. Therefore, any ALM formulation that aims at properly describing the load degradation along the span provoked by end effects must be able

to capture this trace via a proper sampling of the flow field. It has already been demonstrated based on the data reported in Figs. 12, 13 and 14 how the *LineAverage* in its standard settings ($r_s$=1c) misses the tip vortex trace, as the latter is concentrated in a limited region around the airfoil AC.

A simple solution to this issue is shown in Fig. 15 for the low- (*pitch=6°*) and high-load (*pitch=10°*) cases and consists in reducing the sampling radius until the sampling line collapses on the Aerodynamic Centre (AC) line. Including the AC

sampling in the comparison allows for establishing a lower limit on the achievable probing distance and provides a useful comparison with what is currently the most common sampling approach used in ALM codes (Martinez et al., 2012; Sørensen and Shen, 2002; Shamsoddin and Porté-Agel, 2014; Bachant et al., 2018). The effectiveness of reducing the sampling radius is quantified by comparing the angle of attack data from F-ALM simulations with the equivalent 2D angle of attack $\alpha_{2D, eq}$ spanwise profiles from BR-CFD, here used as a benchmark. At the standard set-up of $r_s$=1c, F-ALM yields the same profile

as Fig. 10 for all kernel shapes. The correspondence with the BR-CFD equivalent angle of attack $\alpha_{2D, eq}$ in the region least affected by the tip vortex, i.e., up to ca. 50% of the span, is reasonable, since the airfoil aerodynamic is still dominated by the



undisturbed inflow conditions. In the rest of the span ($0.5 < z/2H < 1$) instead, where the local deformation induced by the tip vortex cannot be neglected, the angle of attack sampled by the F-ALM with the *LineAverage* presents an unphysical trend, opposite to the BR-CFD equivalent angle of attack $\alpha_{2D, eq}$. This aspect has already been discussed in Section 6.2.1. Progressively

reducing the sampling radius, the correspondence between $\alpha$ and $\alpha_{2D, eq}$ worsens in the first half of the blade, while it notably improves in the tip region, especially after $r=\beta=0.1c$. After this point, in fact, the sampled $\alpha$ decreases with height, as would be expected based on physical reasoning. This is also true for the *Gauss-Gumbel* case, but the sampled AoA profile is shifted instead to values higher than the nominal blade pitch, so that no matching between the reference $\alpha_{2D, eq}$ curve and the range of the sampled data is found. Therefore, this function will not be considered for further investigation.

It can be inferred from the considerations made so far that an ideal strategy would increase the sampling distance in the midspan part of the blade, for then progressively reducing it towards the tip. A hint of this fact is also found in the original paper about the *LineAverage* method (Jost et al., 2018), but the authors never developed the concept further. The key aspect to be noticed here though is that for all cases considered in Fig. 15, there is a region in the last 20% of the blade ($z/2H > 0.8$) where the equivalent AoA from BR-CFD falls below the lower limit of the sampled one, i.e., the data probed along the AC line. The

position and extension of this region depends on the type of kernel and on the loading conditions, suggesting that it might be related to the intensity of the forces spread into the domain. It is not a chance, for instance, that this region is bigger for the anisotropic case, which underestimates the intensity of the flow field deformation with respect to BR-CFD and F-ALM *isotropic* (see Figs. 12 and 13). Therefore, a connection exists between the quality of the force spreading and that of the velocity sampling processes, which will be demonstrated in Section 6.2.4.


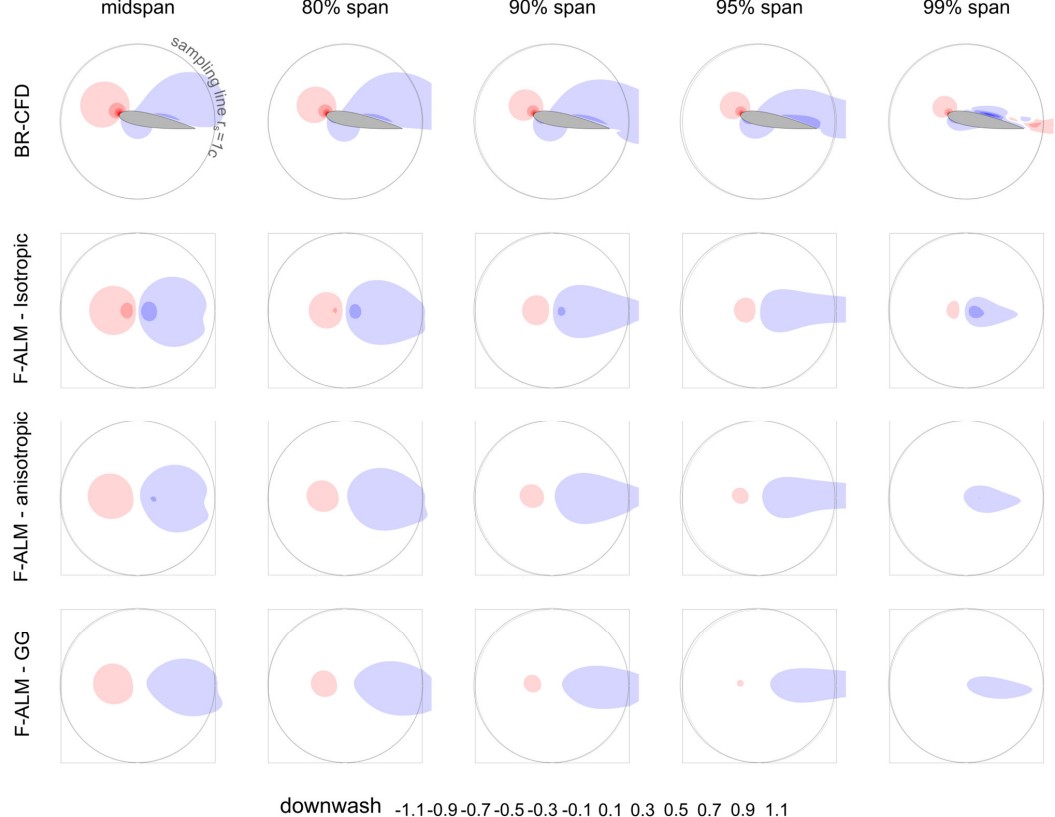

**Figure 14 Comparison in terms of non-dimensional downwash velocity $v_y/V_0$ between BR-CFD and Frozen ALM (F-ALM) at different spanwise sections, for pitch=10°.**

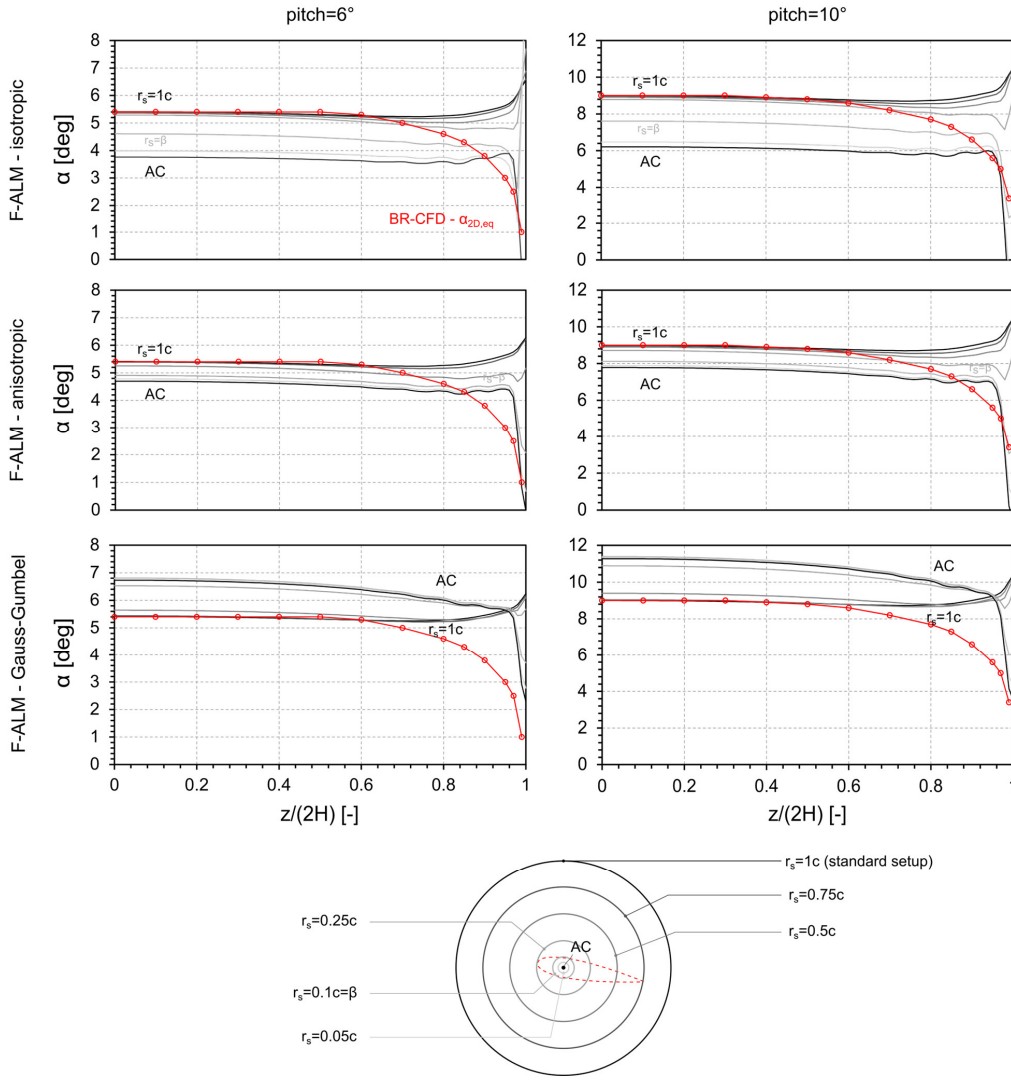

**Figure 15 Comparison in terms of AoA spanwise distribution between BR-CFD, calculated as a dynamically equivalent AoA, and Frozen ALM (F-ALM), sampled via the *LineAverage* method at different sampling distances. The data under the name "AC" refer to the sampling at the aerodynamic center of the airfoil, corresponding to the force insertion point.**

### 6.2.3 Frozen ALM - Tip vortex structure

Figure 16 shows the comparison in terms of non-dimensional downwash velocity $V_y/V_0$ between frozen ALM (F-ALM) and blade-resolved CFD (BR-CFD) for three different load/pitch conditions. The *downwash* velocity is sampled in the near-wake, along a line one chord away from the blade AC.

It is apparent how the F-ALM gives a fair estimation of the downwash immediately downstream of the wing trailing edge (x=1c) up to 80% of the span, for then losing rapidly accuracy in the last 20%. In fact, the magnitude of the velocity induced by the shed vorticity in the tip region is heavily underestimated, consistently with the fact observed in Section 6.2 that F-ALM tends to overspread the computed forces over a wider area compared to high-fidelity simulations. The effect of the kernel shape is not as marked as on the spanwise velocity field (see Fig. 14), although small discrepancies between different functions are still present, especially with the *Gauss-Gumbel* kernel.




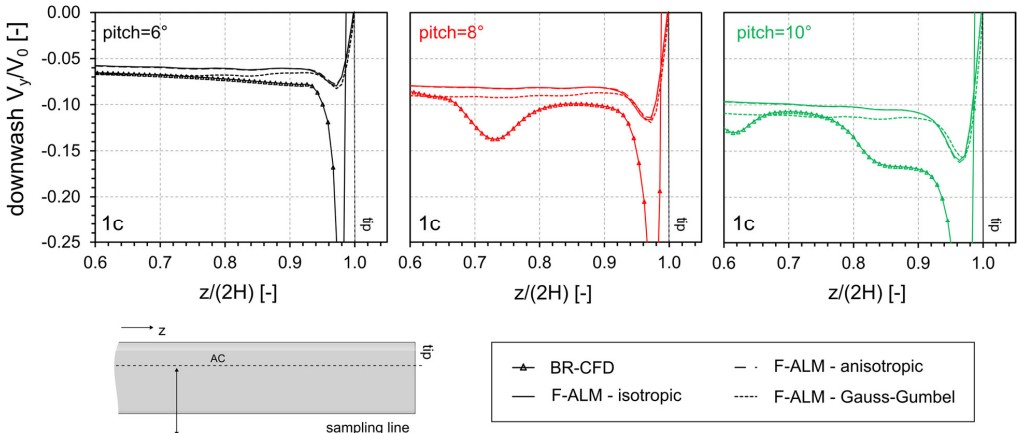

**Figure 16 Comparison in terms of spanwise non-dimensional downwash velocity between BR-CFD and Frozen ALM (F-ALM) for**
**isotropic, anisotropic, and Gauss-Gumbel kernel shapes at the three operating conditions under consideration.**

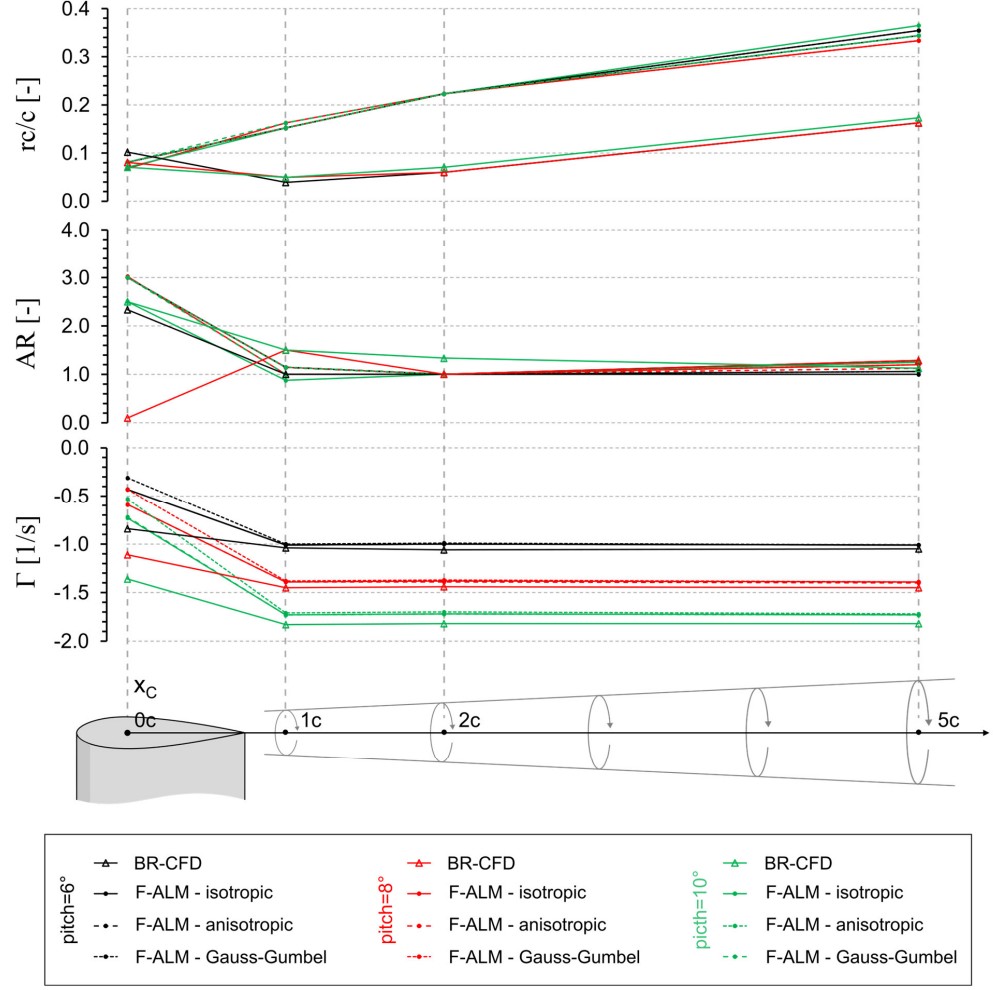

**Figure 17 Comparison in terms of tip vortex core radius, aspect ratio and intensity up to 5 chords downstream of the airfoil between**
**BR-CFD and Frozen ALM, for three different kernel shapes (isotropic, anisotropic, Gauss-Gumbel) and pitch = [6, 8, 10].**



Shifting the focus on the tip vortex structure in the near- and far-wake reported in Fig. 17, the difference between the various
kernel functions becomes negligible, as the flow behaviour is dominated by the integral balance between the bound vorticity along the blade and the one shed into the wake. In fact, the circulation $\Gamma$ of the tip vortex is well predicted by F-ALM, proving that the method is conservative, although a small deviation is observed at the higher loads, e.g., *pitch=10°*. Regarding the shape of the tip vortex, it can be inferred from Fig. 17 that the frozen ALM approach, when the kernel width is properly tuned (see Section 3.1), provides a satisfying estimation of the BR-CFD vortex characteristics, i.e., core radius $r_C$ and Aspect Ratio
AR, especially in the near wake (x/c <=1). In the *far wake*, due to accelerated vortex aging, the $r_C$ value predicted by ALM is overestimated with respect to the BR-CFD one, with a maximum deviation of +100% at x/c=5. In the authors' view, this might be related to the absence in the ALM of the turbulence generation normally occurring in presence of a physical blade tip. A minor role might also be played by the inevitable difference in grid resolution between ALM and BR-CFD.

### 6.2.4 Standard ALM – Spanwise flow

The analysis carried out in Section 6.2.2 established that the Actuator Line Method can, with its own limitations, reproduce the spanwise flow distortion induced by the tip vortex and observed in BR-CFD, and that this effect can be extracted from the flow field if the proper angle of attack sampling strategy is selected. A first-attempt proposal, inferred from the data reported in Fig. 15, is to progressively reduce the sampling radius $r_s$ moving towards the blade tip. The examination of the simulation data also showed that, although the aerodynamic forces are the same of BR-CFD (frozen ALM), the ALM underestimates the
local circulation intensity in the tip region, resulting in: a) lack of intersection between the BR-CFD equivalent angle of attack and the range of the sampled one in the last 20% of the blade (z/2H > 0.8); b) a lower wake downwash in the last 20% of the blade (z/2H > 0.8); c) a more diffused tip vortex structure.

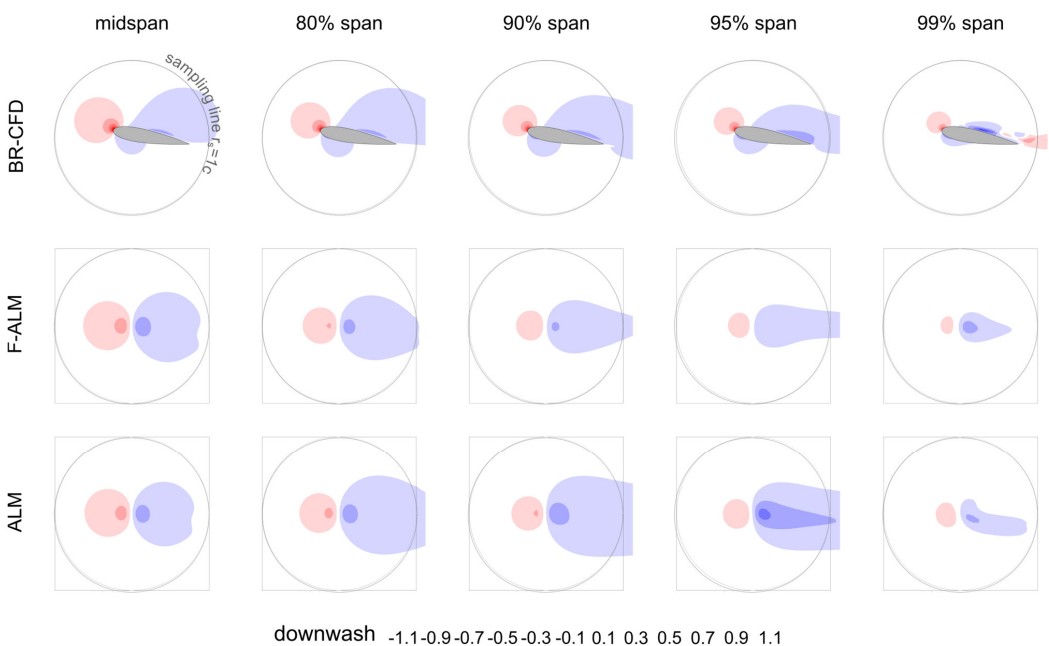

**Figure 18 Comparison in terms of non-dimensional downwash velocity v/V₀ between BR-CFD, Frozen ALM (F-ALM) and standard ALM at different spanwise sections, for pitch=10°. Only the isotropic kernel shape is here considered.**

To provide a further insight into this last issue, the investigation of Section 6.2.2 is carried out again, this time including simulations with the standard ALM method, where the forces are computed from the sampled angle of attack. As it has been

already demonstrated that alternative kernel shapes such as the *anisotropic* and *Gauss-Gumbel* ones, when not reducing the

accuracy of the simulation, provide no additional benefits, only the common *isotropic* function is considered from now on.

Figure 18 reports the comparison in terms of non-dimensional downwash $V_y/V_0$ between BR-CFD, F-ALM and standard ALM for the high-load condition (*pitch=10°*). Only *isotropic* kernel is considered. It is immediately evident how the standard ALM method predicts a flow distortion along the span more similar to that of BR-CFD than the frozen approach. Both the intensity of bound circulation, whose trace can be found in the positive downwash region upstream of the airfoil, and of the

trailing vorticity related to the tip vortex are captured better than in the frozen ALM case. A relevant improvement in the prediction of the lateral extension of the downwash region in the rear part of the wing is achieved.

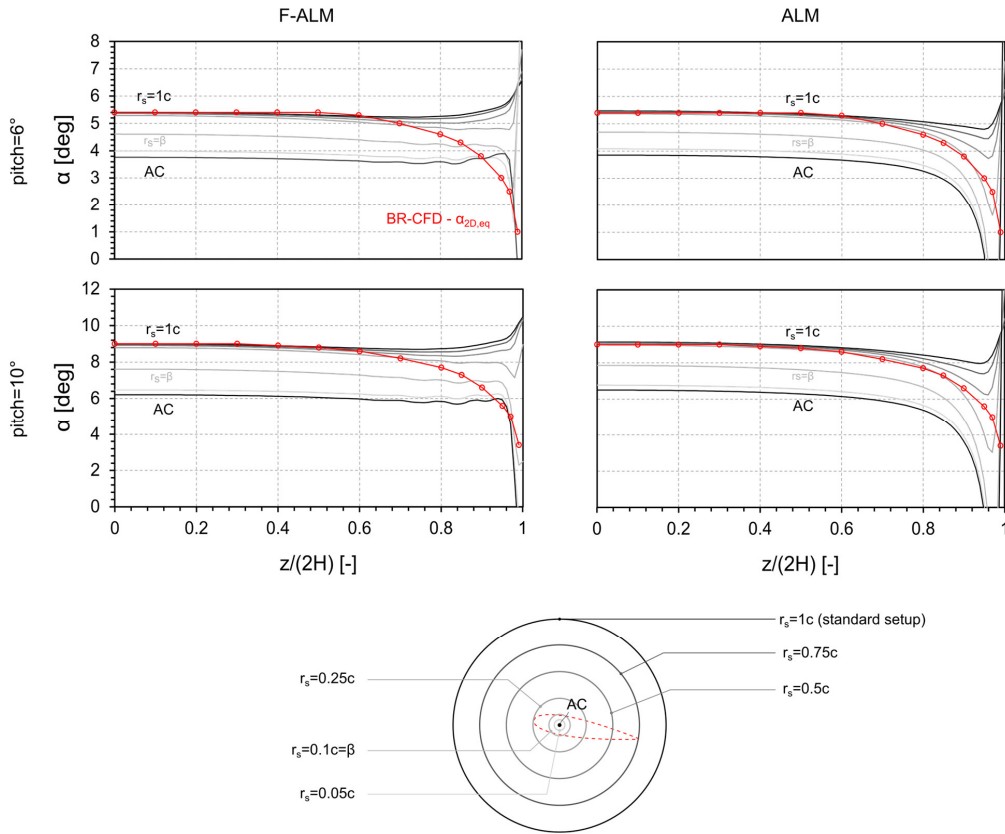

**Figure 19 Comparison in terms of AoA spanwise distribution between BR-CFD, calculated as a dynamically equivalent AoA, Frozen
ALM (F-ALM) and ALM, sampled via the *LineAverage* method at different sampling distances. Only the isotropic kernel shape is
here considered. The data under the name "AC" refer to the sampling at the aerodynamic center of the airfoil, corresponding to the
force insertion point.**

Such an accuracy increase reflects directly on the sampled angle of attack, as visible in Fig. 19. Repeating the experiment carried out in Section 6.2.2 and reported in Fig. 15, it is possible to see indeed how the reference $\alpha_{2D,\ eq}$ from BR-CFD

simulations (see Section 6.1) is now fully contained in the set of curves obtained by sampling the angle of attack along circumferences of decreasing radius $r_s$ with the *LineAverage*. In these conditions, the incompatibility observed in the frozen ALM results (see Fig. 15) is eliminated and the angle of attack information required by the ALM for a correct spanwise load computation can be fully extracted from the flow field. It is interesting to notice that the sampling radius $r_s=0.1c$, which corresponds to the kernel size $\beta=0.1c$ used for the simulations, represents once again a critical threshold, in this case the



minimum sampling radius required to have the complete description of the angle of attack variation along the span. This information will be very important in the future for the formal definition of a novel sampling strategy to be included in the ALM formulation.

The results commented so far may appear unexpected at a first, superficial glance. How is it possible in fact, that the standard ALM, which predicts an unphysical load profile along the span (see Fig. 11), provides a better description of the flow field

around the wing than *frozen ALM*, thus improving the quality of the sampled α data? The answer lies in the tendency of the ALM to "overspread" the forces into the grid, well-known in the scientific community and described in this work in Section 6.2.2. In frozen ALM simulations, as the forces come from BR-CFD and the kernel size is constant along the span, this effect remains uncorrected, resulting in a weaker tip vortex effect. In the standard ALM instead, the same effect is artificially compensated by the increase of the computed aerodynamic forces, and so of the local circulation intensity, towards the tip.

Based on the previous considerations, a strategy to unite the two tendencies, i.e., a correct force distribution along the span and, at the same time, a coherent circulation development towards the tip is, intuitively, to reduce the kernel size with the spanwise position on the wing. This procedure has been already successfully tested by Jha et al. in multiple studies regarding the ALM simulations of horizontal-axis rotors (Jha et al., 2014; Jha and Schmitz, 2018), but so far has not spread too much in the ALM community due to the increased computational cost. As the object of the present work is to provide physics-based

guidelines for the formulation of new sampling/force smearing strategies, rather than reviewing the existing ones or even proposing a new one, this subject will be postponed to a future work.

### 6.2.5 Standard ALM - Tip vortex structure

The improvement in the description of the spanwise flow field provided by the standard ALM positively reflects on the prediction of the tip vortex structure in the wake, as shown in Figs. 20 and 21. Although it extends over a spanwise region (0.9

< z/(2H) < 1) wider than the one observed in F-ALM, the downwash peak predicted by ALM is closer in magnitude to the BR-CFD one, for all loading conditions. Coherently, the predicted tip vortex presents an aging speed, especially in the far wake, more similar to the BR-CFD one, although the inherent limitations in terms of vortex diffusion associated with the ALM approach remain (see Section 6.2.2).

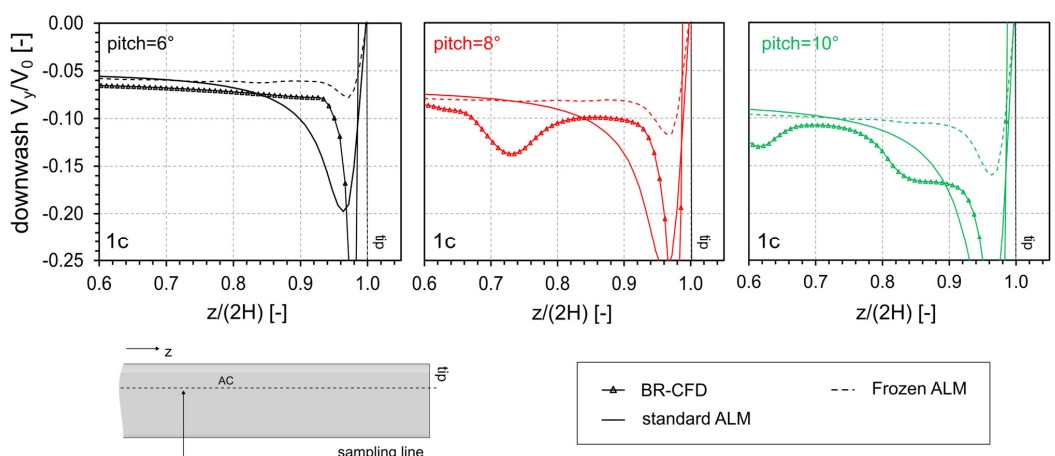


**Figure 20 Comparison in terms of spanwise non-dimensional downwash velocity between BR-CFD, Frozen ALM (F-ALM) and standard ALM for isotropic kernel shape at the three operating conditions under consideration.**





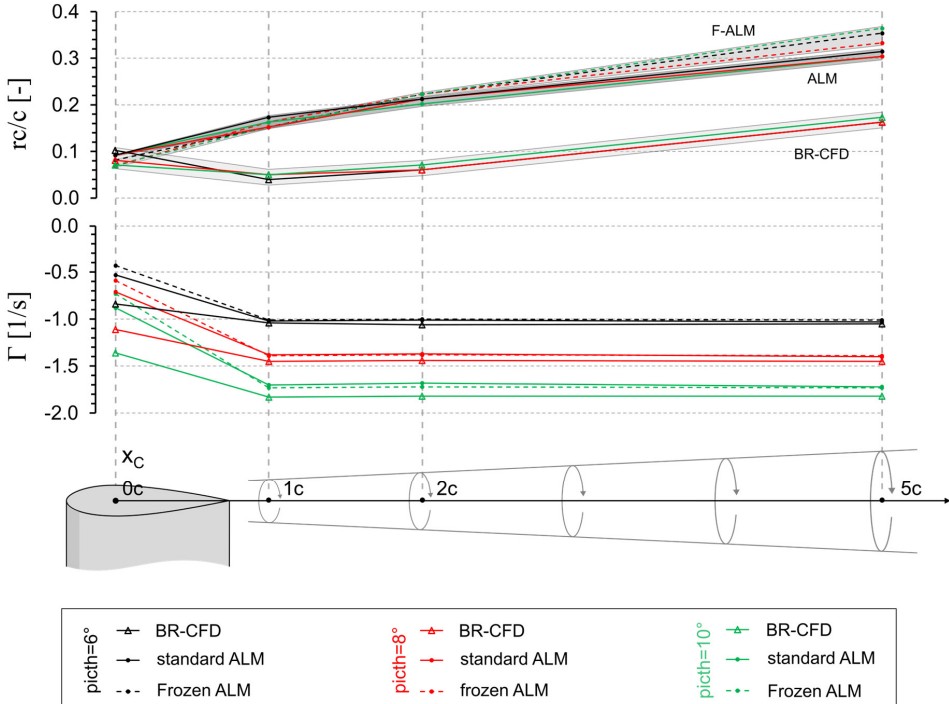

**Figure 21 Comparison in terms of tip vortex core radius and intensity up to 5 chords downstream of the airfoil between BR-CFD, Frozen ALM and standard ALM for the isotropic kernel and pitch = [6, 8, 10].**

## 7 Conclusions

In this study, a comprehensive investigation on the ALM's capability to simulate tip effects has been performed. To this end, a NACA0018 finite wing was the case study for three different simulation techniques: high-fidelity, blade-resolved CFD simulations, to be used as benchmark, standard ALM without any correction and ALM with the spanwise force distribution extracted from blade-resolved data (*frozen ALM*). The analysis has been repeated for three different kernel shapes, *isotropic Gaussian*, *anisotropic Gaussian* and *Gauss-Gumbel*, respectively. For the post-processing and comparison of the data, advanced Vortex Identification Methods for outlining the structure and decay of the tip vortex have been combined with the *LineAverage* technique for the sampling of the local angle of attack along the blade span.

Upon examination of the results, the following conclusions can be drawn:

- until the flow becomes fully three-dimensional – in this case, until the last 5% of the span – the spanwise load degradation observed in BR-CFD can be interpreted as a dynamically equivalent reduction of the local angle of attack. This phenomenon is confined at the blade chord scale;

- the scale of interaction reduces when moving from a region dominated by bound circulation, such as the blade midspan, to one dominated by trailing vorticity such as the tip. Therefore, when modelling tip effects in the ALM framework, it is key that the characteristic length of both force smearing and angle of attack sampling from the flow field decrease approaching the blade extremity. Numerical details of this procedure are out of the scope of the present paper and will be detailed in a future work;

- the ALM produces a more diffused vortex than BR-CFD. Correspondingly, the vortex intensity and downwash in the wake are underestimated. This deviation increases with the blade loading, i.e., pitch angle. The vortex aging on the other hand is overestimated, especially in the far-wake (5 chord away from the wing tip);



- the use of alternative kernel shapes (sometimes proposed as a countermeasure in the literature) such as *anisotropic Gaussian* or *Gauss-Gumbel* does not introduce relevant differences in the predicted loads and tip-vortex structure but lower the capability of the ALM to extract the proper angle of attack from the flow field. Therefore, their use is not recommended.

## Acknowledgments

This study is the expanded version of a preliminary one presented at the WESC 2023 conference by the European Academy of Wind Energy (EAWE). The authors would like to acknowledge Prof. Giovanni Ferrara and Dr. Francesco Papi from Università degli Studi di Firenze, for supporting this activity and for the useful technical discussions.

## Financial support

This work did not receive external financial support. However, it has exploited a 20% fee reduction thanks to the contribution of the European Academy of Wind Energy.

## Competing interests

One of the authors is a member of the editorial board of *Wind Energy Science*. The peer-review process was guided by an independent editor. The authors have no other competing interests to declare except what is implied by their affiliation.

## Author contribution

PFM conceptualized the work, carried out ALM and CFD analyses, and was responsible for the first draft preparation. OS and SC helped in the ALM analyses and Vortex Identification Methods, respectively. FB and AB helped in orientating the research, critically discussing the analysis, and reviewing the paper. AB coordinated the research team.

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
