# Peer review of "An insight into the capability of the Actuator Line Method to resolve tip vortices"

_Wind Energy Science, 2023_

## Referee Comment (RC1)

*Comments to the paper no.* **wes-2023-88**

**An insight into the capability of the Actuator Line Method to resolve tip vortices**

The authors present a very interesting, detailed and well designed numerical analysis to investigate the origin of ALM limitations (if any) in describing blade tip angle of attack and loads behavior. To this aim a very simple straight 3D wing geometry has been selected and many different analyses dealing with AOA, loads, velocity field, tip vortex structure are suitably addressed.

Moreover, a very important effort is done by the authors to derive some general guidelines on the use of ALM-based solvers on the basis of their numerical evidences.

The paper is well written, although there are some minor technical corrections that I have proposed in the following. I also appreciate the writing style which, except for a fee parts (see comments below), suitably follows the investigations conducted keeping the reader always focused.

Generally, in the paper both present and past tenses are used when referring to literature results and also presented results: please make a coherent choice. I strongly suggest to use always the present tense.

As a general comment I found the paper a bit long. I understand that the investigation is wide and it requires suitable comments and descriptions. Nevertheless, I encourage the authors to try to make the paper more concise also trying to avoid some repeated or too much "educational" sentences. Some examples are reported in the following.

My indication is to **ACCEPT** the paper only after **MINOR REVISIONS** following the comments listed below.

**SPECIFIC COMMENTS**

Abstract
l. 10: the paper is focused on the ability of the ALM to describe blade tip vortex flow features. Which other vortex-like structures do the authors refer to in the abstract? The same sentence is repeated in the Introduction (l. 36). Please rephrase these sentences to clarify.

Introduction
1. The introduction is well written and correctly reports the investigations on the limitations and capabilities of ALM solvers to describe wind turbine (WT) blades aerodynamic behaviour. Lines 24-35 outline an overview of the existing methods for WT rotor aerodynamics modelling. Within this general summary, which span from engineering-type BEMT formulations up to CFD, mid-fidelity vortex methods should be mentioned. Indeed, this family of 3D solvers is well-known to be able to capture (at a very reduced computational cost) rotor blade aeroloads over a wide range of operating conditions and it has been demonstrated that they provide an accurate prediction of blade wake shape close to the rotor disk. The latter aspect is particularly related to the topic discussed in the paper and, for instance, in a future work it would be very interesting to compare the ALM strategy proposed by the authors to the outcomes of such models. Some examples of effectiveness of the mentioned methods are reported in the following papers and I suggest to mention them for completeness as alternative solvers able to

describe tip vortices features. The second work, in particular, includes specific computations of wind turbine blade wake shape and velocity field downstream.

Boorsma, K., et al, "Progress in the validation of rotor aerodynamic codes using field data", Wind Energy Science, 2023, 8(2), pp. 211–230.

Greco, L., Testa, C., "Wind turbine unsteady aerodynamics and performance by a free-wake panel method", Renewable Energy, 2021, 164, pp. 444-459.

2. l. 25: replace "framework" with "domain".
3. l. 27: "required for their aeroelastic…"
4. l. 46: ".. of the ALM element size h_ALM < 0.25beta…": the symbol h_ALM is not defined in the text and it is not clear, at this point, what does "ALM element" refer to and which size the authors are considering. Please rephrase the sentence to better clarify.
5. Section 1.2: the standard ALM is well-known to be an iterative method. Presumably also the Frozen-ALM requires an iteration, but this is not explicitly mentioned. Moreover, the "frozen" definition does not exactly suggest that there is an iteration. I suggest the authors to better clarify this aspect.

Section 2

The description of the test case must be placed within the numerical results section and not before the description of the numerical methods.

On lines 94-95 the authors state that the Reynolds number was selected to obtain a linear behaviour of the airfoil. It is not clear which type of linearity the authors refer to. Is it referred to the airfoil lift? Please clarify.

Section 3

1. In the description of the ALM the AOA calculation step is not mentioned.
2. Line 104: replace "rotor" with "wing".
3. Line 105: replace "...then, based on the sampled flow field…" with "...then, based on this…"
4. Line 106: replace "projected" with "exerted as sources of momentum"
5. Line 112: it has been already mentioned that the ALM relies on tabulated airfoil polar data.
6. Line 123: replace "at which" with "where".
7. Eq. 2: symbols rc and rt are not defined in the text.
8. Line 158: "A uniform cartesian grid…": please clarify if this refers to the spanwise direction or to a different one.
9. Line 159: the symbol h_ALM is not defined in the text.
10. Line 161: replace "The anisotropic and Gauss…." with "Differently, the anisotropic…."
11. Line 163: replace "As this process took place…" with "As this process does not consider….".
12. Table 3: the columns of ALM-iso and ALM-GG show only one value, it is not clear why.

Section 5

1. Line 239: the vortex Aspect Ratio is not defined in the text. Please revise.
2. Section 5.2: Equation 9 is not clear to me. Does the vector sj indicate the tangent-to-the-line unit vector? If so, then symbol |sj| denotes its magnitude, hence the dot product with the velocity vector is not needed. Moreover, on line 258 only the trailing vorticity is mentioned whilst the proposed method can, within a potential flow assumption, take into account both trailed and shed vorticity. Please revise this part.

Section 6

1. Line 273: the Frozen ALM has been previously defined, no need to repeat here.

2. Lines 287 – 290: the authors state that for the BR-CFD computation it is not possible to use the LineAverage method to sample the velocity field due to possible intersection of the sampling line with the airfoil. Indeed, if the sampling circle is centered at c/4 it would be possible to sample the velocity on a circle with r greater than ¾ c. This type of velocity sampling would be more consistent with respect to some of the proposed ALM computations and would eliminate the need of an additional 2D CFD computation.
3. Line 305: "...section lift computed at alpha …"
4. Lines 306-308: "This effect is also….SS suction peak". This sentence is not clear. The reduction of the SS peak is evident from Fig. 7 but it is not clear to which shift of the stagnation point the authors refer to. Please clarify.
5. Line 310: "… 2D BR-CFD towards the trailing edge"
6. Line 313: "… and 8d-e (bottom)"
7. Line 315: "… the corresponding blade section takes the name …"
8. Line 320: "Differently, in the region between 97% of the span and the tip  the flow…"
9. Line 322: "… in Fig.9 (right)".
10. Line 337: "… Section 6.1 demonstrates …"
11. Line 338-339: Please remove "along the blade". It is already clear that the progressive reduction of AOA_eq refer to the spanwise direction.
12. Line 341-342: Please remove "if the answer is positive".
13. Section 6.2.1: From my understanding, the AOA shown in Fig. 10 and computed from the BR-CFD simulation is the outcome of the 2D-equivalence (so it is alpha_2D_eq). Is it correct? In this case, it is not surprising that this AOA (coming from an equivalence which is not providing accurate results at the tip) is not consistent with the loads from BR-CFD shown in Fig. 11. From this point of view (and also considering the conclusions of the paper) I think it is worthy computing the AOA from BR-CFD using the LineAve methodology to better clarify these aspects. Moreover, in Fig. 11 the results of the F-ALM are not shown (or are they coincident with the standard ALM?).
14. Section 6.2.2: **Lines 408-410** can be shortened for the sake of brevity. In lines 411-413 the authors comment that the BR-CFD and F-ALM predicted velocity fields are similar. In my understanding the mentioned symmetric behaviour refers to the chordwise direction but it is never indicated in the text. Moreover, the predicted fields do not show the rotation of the velocity field about the z axis (induced by the presence of the airfoil), which should be a well known limitation of ALM approaches. Finally, the BR-CFD downwash velocity shows an asymmetric chordwise behaviour already at the mid-span section. The authors should comment more in deep these different field features and link them to the actual limitations of the ALM approach. **Line 417:** please remove "...commented in the previous paragraph" and add the reference to the actual section. **Lines 418-420**: these lines refer to evidences of Fig.17 that, at this point of the paper, has not been mentioned yet. I suggest to include these comments where Fig 17 is commented or, alternatively, move Fig 17 in this section.
15. Line 441: "...airfoil aerodynamics…"
16. Line 442: "… conditions. As shown in Section 6.2.1, in the rest of the span (0.5<z/2H<1)  …". Please remove "This aspect has already been discussed in Section 6.2.1".
17. Line 446: It is not clear what the author refer to by "height".
18. Section 6.2.4 is directly linked to Section 6.2.2. Thus I suggest to postpone Section 6.2.3 and place those two sections one after each other or even merge them.
19. Line 474: the authors mention the "shed vorticity" which is zero in a steady flow. Please clarify.
20. Figure 17: It is quite hard to distinguish the different results shown in the figure.
21. Line 500: I don't understand the sentence "lack of intersection between the BR-CFD…..". Please rephrase it for better clarity.

22. Line 512: "Only isotropic kernel is considered". It has been already pointed out, please remove.
23. Line 533: Please remove "superficial glance".
24. Lines 535 – 539: these part is not clear. Please rephrase it to better clarify the interpretation of the presented results.
25. After their in-depth analysis the authors come to the conclusion that the standard ALM, if properly tuned (e.g. appropriately selecting the value of rs) can provide accurate predictions of the AOA of the tested wing. It would be very interesting to see the spanwise loads estimate by using the rs value providing the best match of the AOA with respect to the BR-CFD outcomes. I suggest to add to Fig.19 a similar one regarding blade loads.

---

## Author Comment (AC1)

**RESPONSES TO REVIEWER #1**

The authors present a very interesting, detailed and well designed numerical analysis to investigate the origin of ALM limitations (if any) in describing blade tip angle of attack and loads behavior. To this aim a very simple straight 3D wing geometry has been selected and many different analyses dealing with AOA, loads, velocity field, tip vortex structure are suitably addressed. Moreover, a very important effort is done by the authors to derive some general guidelines on the use of ALM-based solvers on the basis of their numerical evidences.

The paper is well written, although there are some minor technical corrections that I have proposed in the following. I also appreciate the writing style which, except for a few parts (see comments below), suitably follows the investigations conducted keeping the reader always focused.

*The Authors would like to thank the Reviewer for his appreciation of the study. The Authors tried their best to improve the paper according to the Reviewer's comments.*

Generally, in the paper both present and past tenses are used when referring to literature results and also presented results: please make a coherent choice. I strongly suggest to use always the present tense.

*Thank you for pointing this out. The present tense has been made uniform throughout the paper.*

As a general comment I found the paper a bit long. I understand that the investigation is wide and it requires suitable comments and descriptions. Nevertheless, I encourage the authors to try to make the paper more concise also trying to avoid some repeated or too much "educational" sentences. Some examples are reported in the following.

*Based on the comments of all Reviewers, the paper has been significantly shortened to avoid redundancies, although new information and data have been added.*

**My indication is to ACCEPT the paper only after MINOR REVISIONS following the comments listed below.**

**SPECIFIC COMMENTS**

Abstract

l. The paper is focused on the ability of the ALM to describe blade tip vortex flow features. Which other vortex-like structures do the authors refer to in the abstract? The same sentence is repeated in the Introduction (l. 36). Please rephrase these sentences to clarify.

*Thank you for the comment. The denomination "vortex-like structures" was intended as an alternative to "tip vortices". As you pointed out, however, this could be misleading. The corresponding lines have been rewritten for clarity.*

Introduction

1. The introduction is well written and correctly reports the investigations on the limitations and capabilities of ALM solvers to describe wind turbine (WT) blades aerodynamic behaviour. Lines 24-35 outline an overview of the existing methods for WT rotor aerodynamics modelling. Within this general summary, which span from engineering-type BEMT formulations up to CFD, mid-fidelity vortex methods should be mentioned. Indeed, this family of 3D solvers is well-known to be able to capture (at a very reduced computational cost) rotor blade aeroloads over a wide range of operating conditions and it has been demonstrated that they provide an accurate prediction of blade wake shape close to the rotor disk. The latter aspect is particularly related to the topic discussed in the paper and, for instance, in a future work it would be very interesting to compare the ALM strategy proposed by the authors to the outcomes of such models. Some examples of effectiveness of the mentioned methods are reported in the following papers and I suggest to mention them for completeness as alternative solvers able to describe tip vortices features. The second work, in particular, includes specific computations of wind turbine blade wake shape and velocity field downstream.

a. Boorsma, K., et al, "Progress in the validation of rotor aerodynamic codes using field data", Wind Energy Science, 2023, 8(2), pp. 211–230.

b. Greco, L., Testa, C., "Wind turbine unsteady aerodynamics and performance by a free-wake panel method", Renewable Energy, 2021, 164, pp. 444-459.

Thank you for these interesting suggestions. The introduction has been largely rewritten to include the recommended information.

2. l. 25: replace "framework" with "domain".
3. l. 27: "required for their aeroelastic..."
4. l. 46: ".. of the ALM element size h_ALM < 0.25beta...": the symbol h_ALM is not defined in the text and it is not clear, at this point, what does "ALM element" refer to and which size the authors are considering. Please rephrase the sentence to better clarify.

All changes have been implemented.

5. Section 1.2: the standard ALM is well-known to be an iterative method. Presumably also the Frozen-ALM requires an iteration, but this is not explicitly mentioned. Moreover, the "frozen" definition does not exactly suggest that there is an iteration. I suggest the authors to better clarify this aspect.

The Reviewer is right. The standard ALM is iterative due to the coupling between the aerodynamic force definition and the flow field computation, although in our implementation this aspect is removed by reducing the timestep accordingly. In the frozen ALM instead, since the forces are an input and not a solver variable, there is no iteration. This aspect has been clarified in the description of the method in Section 3.

Section 2

The description of the test case must be placed within the numerical results section and not before the description of the numerical methods. On lines 94-95 the authors state that the Reynolds number was selected to obtain a linear behaviour of the airfoil. It is not clear which type of linearity the authors refer to. Is it referred to the airfoil lift? Please clarify.

The Reviewer comment is again on point. Section 2 has become Section 6.1 and the type of linearity involved in the test case selection has been clarified. The validation of BR-CFD computations has been moved to a dedicated section under the results to maintain the coherence of the narration.

Section 3

1. In the description of the ALM the AOA calculation step is not mentioned.

The Reviewer is right. The corresponding paragraph has been rewritten to clarify this step.

2. Line 104: replace "rotor" with "wing".
3. Line 105: replace "...then, based on the sampled flow field..." with "...then, based on this...".
4. Line 106: replace "projected" with "exerted as sources of momentum"
5. Line 112: it has been already mentioned that the ALM relies on tabulated airfoil polar data.
6. Line 123: replace "at which" with "where".

Changes have been implemented.

7. Eq. 2: symbols rc and rt are not defined in the text.

Thank you for noting. The corresponding paragraph and Figure 1 have been updated to clarify this notation.

8. Line 158: "A uniform cartesian grid...": please clarify if this refers to the spanwise direction or to a different one.
9. Line 159: the symbol h_ALM is not defined in the text.

The corresponding lines have been revised.

10. Line 161: replace "The anisotropic and Gauss...." with "Differently, the anisotropic....".
11. Line 163: replace "As this process took place..." with "As this process does not consider....".

Changes have been implemented.

12. Table 3: the columns of ALM-iso and ALM-GG show only one value, it is not clear why.
*Table 1 has been revised for clarity.*

Section 5

1. Line 239: the vortex Aspect Ratio is not defined in the text. Please revise.
*Noted. The Aspect Ratio formal definition has been added to the revised text.*

2. Section 5.2: Equation 9 is not clear to me. Does the vector sj indicate the tangent-to-the-line unit vector? If so, then symbol |sj| denotes its magnitude, hence the dot product with the velocity vector is not needed. Moreover, on line 258 only the trailing vorticity is mentioned whilst the proposed method can, within a potential flow assumption, take into account both trailed and shed vorticity. Please revise this part.
*Thank you for the comment. Eq. 9 has been corrected to avoid misinterpretation. The LineAverage accounts indeed for shed vorticity. This aspect has been pointed out in the revised version.*

Section 6

1. Line 273: the Frozen ALM has been previously defined, no need to repeat here.
*Done.*

2. Lines 287 – 290: the authors state that for the BR-CFD computation it is not possible to use the LineAverage method to sample the velocity field due to possible intersection of the sampling line with the airfoil. Indeed, if the sampling circle is centered at c/4 it would be possible to sample the velocity on a circle with r greater than 3⁄4 c. This type of velocity sampling would be more consistent with respect to some of the proposed ALM computations and would eliminate the need of an additional 2D CFD computation.
*Comments are very pertinent. Using the LineAverage for BR-CFD simulations is indeed possible with a sampling radius bigger than 3/4c and has been done to obtain the angle of attack profile reported in Fig. 10. For clarity this aspect has been clarified in the revised manuscript.*

3. Line 305: "...section lift computed at alpha ..."
*Done, thank you.*

4. Lines 306-308: "This effect is also....SS suction peak". This sentence is not clear. The reduction of the SS peak is evident from Fig. 7 but it is not clear to which shift of the stagnation point the authors refer to. Please clarify.
*The Reviewer is right. The shift of the stagnation point is present in Figure 7 but barely visible due to the small magnitude of the AoA involved. The corresponding paragraph has been rewritten to highlight this aspect in a clearer way.*

5. Line 310: "... 2D BR-CFD towards the trailing edge"
6. Line 313: "... and 8d-e (bottom)"
7. Line 315: "... the corresponding blade section takes the name ..."
8. Line 320: "Differently, in the region between 97% of the span and the tip instead the flow..."
9. Line 322: "... in Fig.9 (right)".
10. Line 337: "... Section 6.1 demonstrates ..."
11. Line 338-339: Please remove "along the blade". It is already clear that the progressive reduction
12. of AOA_eq refer to the spanwise direction.
13. Line 341-342: Please remove "if the answer is positive".
*Thank you for noting. All corrections have been implemented.*

14. Section 6.2.1: From my understanding, the AOA shown in Fig. 10 and computed from the BR-CFD simulation is the outcome of the 2D-equivalence (so it is alpha_2D_eq). Is it correct? In this case, it is not surprising that this AOA (coming from an equivalence which is not providing accurate results at the tip) is not consistent with the loads from BR-CFD shown in Fig. 11. From this point of view (and

also considering the conclusions of the paper) I think it is worthy computing the AOA from BR-CFD using the LineAve methodology to better clarify these aspects. Moreover, in Fig. 11 the results of the F-ALM are not shown (or are they coincident with the standard ALM?).

Thank you for the comment. The angle of attack profile reported in Fig. 10 for BR-CFD has been sampled from the flow field using the LineAverage method. In Fig. 11, the loads of F-ALM coincide with the BR-CFD ones since they are taken from there. For clarity, these aspects have been clarified in the revised manuscript.

15. Section 6.2.2: Lines 408-410 can be shortened for the sake of brevity.

Done. Thank you.

16. In lines 411-413 the authors comment that the BR-CFD and F-ALM predicted velocity fields are similar. In my understanding the mentioned symmetric behaviour refers to the chordwise direction, but it is never indicated in the text. Moreover, the predicted fields do not show the rotation of the velocity field about the z axis (induced by the presence of the airfoil), which should be a well-known limitation of ALM approaches. Finally, the BR-CFD downwash velocity shows an asymmetric chordwise behaviour already at the mid-span section. The authors should comment more in deep these different field features and link them to the actual limitations of the ALM approach.

The Reviewer's comment is pertinent. The paragraph was not clear in the definition of the cited "symmetric behavior", which indeed was referred to the size of the upwash and downwash regions using the airfoil chord normal direction as a reference. This part has been completely re-written to clarify this aspect.

17. Line 417: please remove "...commented in the previous paragraph" and add the reference to the actual section.

Done, thank you.

18. Lines 418-420: these lines refer to evidences of Fig.17 that, at this point of the paper, has not been mentioned yet. I suggest including these comments where Fig 17 is commented or, alternatively, move Fig 17 in this section.

These lines have been moved to where Fig. 17 is commented.

19. Line 441: "...airfoil aerodynamics..."
20. Line 442: "... conditions. As shown in Section 6.2.1, in the rest of the span (0.5<z/2H<1)
21. instead ...". Please remove "This aspect has already been discussed in Section 6.2.1".
22. Line 446: It is not clear what the author refer to by "height".

All changes have been implemented.

23. Section 6.2.4 is directly linked to Section 6.2.2. Thus I suggest to postpone Section 6.2.3 and place those two sections one after each other or even merge them.

Thank you for the suggestion. The result structure has been re-organized accordingly by merging the section concerning spanwise flow and tip vortex structure.

24. Line 474: the authors mention the "shed vorticity" which is zero in a steady flow. Please clarify.

Done.

25. Figure 17: It is quite hard to distinguish the different results shown in the figure.

Thank you for the comment. Fig. 17 has been revised to improve readability and highlight the important trends.

26. Line 500: I don't understand the sentence "lack of intersection between the BR-CFD.....". Please rephrase it for better clarity.

In the revised manuscript, the corresponding paragraph has been deleted.

27. Line 512: "Only isotropic kernel is considered". It has been already pointed out, please remove.
28. Line 533: Please remove "superficial glance".

Changes have been implemented.

29. Lines 535 – 539: this part is not clear. Please rephrase it to better clarify the interpretation of

the presented results.

Agreed. The corresponding part has been re-written for clarity.

30. After their in-depth analysis the authors come to the conclusion that the standard ALM, if properly tuned (e.g. appropriately selecting the value of rs) can provide accurate predictions of the AOA of the tested wing. It would be very interesting to see the spanwise loads estimate by using the rs value providing the best match of the AOA with respect to the BR-CFD outcomes. I suggest adding to Fig.19 a similar one regarding blade loads.

Thank you for the suggestion. A new Figure has been added to the revised manuscript, showing the comparison in terms of blade loads between BR-CFD and ALM with the standard sampling radius and the optimal one coming from this investigation.

---

## Author Comment (AC2)

**RESPONSES TO REVIEWER #2**

**General comments**

Not being familiar with the ALM subject I thought I a very difficult review. I am happy to report that despite the novelty of the subject (for me), the paper allowed me to understand a great deal of the overall subject and the authors interest.

Overall, the paper is well written with useful plots illustrating the points the authors wanted to make. Despite some minor syntax errors mostly attributed to the paper's length the text can be read easily. Being unexperienced in the topic, having sections dedicated to the state of the art and explanations about the theory was very useful, although it may be redundant for more experienced readers. Having such sections in appendices could have been another option.
The Authors would like to thank the Reviewer for his appreciation of our study and tried their best to further improve it based on the comments received. Regarding the theoretical bases, the authors would be inclined to leave the corresponding sections within the text since they are thought to be instrumental for the understanding of the numerical approached proposed herein.

The comments and questions below are for my clarification. I believe that the paper should be **accepted** after the **minor comments** in the questions below are answered.

**Technical comments**
1. I understand the need of a well-known and simple airfoil such as NACA 0018, despite no longer being used in wind turbine designs. However, the Reynolds number is too low to be representative of modern wind turbine, some viscous effect may still be seen thereby possibly questioning the outcome. Such study would have been preferential above Re = 2-3 10^6. Did you choose 500k in order to save some computing time?
2. The turbulent intensity seems quite high compared to wind tunnel experiment or even some wind farm sites following the IEC classification. Why did you choose 1 % and not a lower value (which could have helped your CFD simulation)?

Thank you for the comments. An intermediate Reynolds number of 500e3 has been selected as an optimal compromise between computational cost and achievable accuracy. In our experience, although viscous effects are still relevant, RANS solvers perform reasonably well in such operating conditions. The main drawback is the need of a turbulence model accounting for laminar-to-turbulent transition – in this case, the k-ω SST with intermittency transport – that also requires higher levels of inlet turbulence intensity to work in a stable way. This is the reason for the selected value of 1% for the TI. We have clarified these choices in the revised manuscript.

3. The CFD setup followed a previous work performed by the authors, the RANS approach tends to smooth out any flow unsteadiness and presents only averaged results. When looking at the unsteady phenomena of tip vortices, another approach would have been more pertinent such as LES or (i)DDES even URANS. Did you perform sensitivity studies on the CFD baseline as it could impact the rest of the study (such as the AoA recognition)?
4. Similarly, the vortex detection may not be fully accurate using a RANS approach rather than a time varying one.
5. Figure 7 and Figure 8 are very clear highlighting the three-dimensionality effect introduced by the finite wing. Because of the RANS solver 3D cases may not be truly those plotted, or at least the error bar due to the unsteady nature cannot be seen.

Reviewer's comments are pertinent. The set-up of the numerical simulations has been dictated by the main scope of the paper, which is not an exact reproduction of experimental measurements, but rather the investigation of the main mechanisms behind tip losses effects using a high-fidelity method (BR-CFD) and of the ALM capability of reproducing such effects at parity of other modeling choices. In this perspective, flow features that would not be captured anyway by the ALM formulation, mainly unsteadiness in the 3D flow field across the wing and in the tip vortex shedding, have been removed on purpose.
The reliability of the selected post-processing techniques on the hand has been verified in previous works by the authors, at least for the quantities of interest for the present investigation:

- *Tip vortex tracking:* the vortex identification methods used in this work has been validated against unsteady measurement and numerical simulations of a wind turbine in a previous work of one of the authors (https://doi.org/10.5194/wes-8-1659-2023). This study has also shown that a RANS methodology can correctly predict the average metrics of the shed tip vortices, despite their inherently non-stationary nature, with limited differences from an LES approach. For this reason, the inaccuracies introduced by performing a RANS simulation rather than LES can be considered limited in terms of tip vortex convection velocity, core radius and strength;
- *LineAverage:* this methodology has been applied by its creators (https://doi.org/10.1002/we.2196) and by the authors (https://doi.org/10.1016/j.enconman.2020.113284) in previous works on high-fidelity simulations of both horizontal- and vertical-axis wind turbines, showing accurate results in both steady and unsteady conditions.

6. Very impressive 2D CFD results matching the experimental CL and CD!
7. Did you average the whole wing to produce the CL and CD curve for the 3D BR-CFD (fig 4)? It could have been interesting to plot some of the different positions along the span as used further down the paper to illustrate the spread in results.

The 3D BR-CFD data was computed upon integration of the local pressure over the wing surface. The text has been updated to clarify this aspect. Furthermore, following your suggestion, the corresponding figure has been updated with the lift and drag curves computed at different spanwise positions along the wing.

8. To complete those interesting figures (Figure 7 and Figure 8), it may be useful to add onto the wing shape the 3D AoA contour (if feasible)? It could show interesting pattern regarding its spanwise evolution.
9. The evolution of the radial flow component could also be plotted onto the wing surface since its behaviour is monitored in the next sections.

Thank you for the interesting suggestion. We believe that the information in terms of flow deformation induced by the tip vortex can be presented by 3D streamlines as effectively as the suggested plotting variables (3D AoA, radial flow component). Figures 7 and 8 have been updated accordingly.

10. Did the 3D AoA calculations account for the radial flow in the Cp calculation? The equation 9 mention the local velocity, is the possible "flow deviation" due to the tip accounted for?

Thank you for noting. The *LineAverage* method only accounts for the velocity components in the same plane of the sampling line, complying with the conventional definition of the angle of attack. This aspect has been clarified in the text. Based on the method's formulation, it is then supposed to account for any flow deviations locally induced by the tip vortex.

11. The sentence in Line 389-394 (However, the magnitude…) is very important and should be split into several sentences to avoid losing the reader.

Agreed. The corresponding sentence has been rewritten.

12. The conclusion is clear and well written summarizing the outcome and future work. Will the bullet point 2 solve/address the issue in bullet point 3?

This is an interesting question. It is not clear from the study whether the gap in terms of tip vortex structure between ALM and BR-CFD can be filled by improving the ALM formulation or it is related to the intrinsic differences between the two methods. Our opinion is that, even it was possible, the corresponding mesh requirements would be so high to make the ALM not convenient anymore. The conclusions of the paper have been updated to highlight this aspect.